# The NASA Carbon Airborne Flux Experiment (CARAFE): Instrumentation and Methodology

Glenn M. Wolfe[1,2], S. Randy Kawa[1], Thomas F. Hanisco[1], Reem A. Hannun[1,2],  Paul A. Newman[1], Andrew Swanson[1,3], Steve Bailey[1], John Barrick[4], K. Lee Thornhill[4], Glenn Diskin[4], Josh DiGangi[4], John B. Nowak[4], Carl Sorenson[5], Geoffrey Bland[6], James K. Yungel[7], and Craig A. Swenson[8]

[1]Atmospheric Chemistry and Dynamics Lab, NASA Goddard Space Flight Center, Greenbelt, MD, USA
[2]Joint Center for Earth Systems Technology, University of Maryland Baltimore County, Baltimore, MD, USA
[3]Goddard Earth Sciences Technology and Research, Universities Space Research Association, Columbia, MD, USA
[4]NASA Langley Research Center, Hampton, VA, USA
[5]NASA Ames Research Center, Moffett Field, CA, USA
[6]Wallops Flight Facility, NASA Goddard Space Flight Center, Wallops Island, VA, USA
[7]AECOM,  ATM Project,  Wallops Flight Facility,  NASA Goddard Space Flight Center, Wallops Island, VA, USA
[8]SSAI, ATM Project, Wallops Flight Facility, NASA Goddard Space Flight Center, Wallops Island, VA, USA

*Correspondence to*: Glenn M. Wolfe (glenn.m.wolfe@nasa.gov)

**Abstract.** The exchange of trace gases between the Earth's surface and atmosphere strongly influences atmospheric composition. Airborne eddy covariance can quantify surface fluxes at local to regional scales (1 – 1000 km), potentially helping to bridge gaps between top-down and bottom-up flux estimates and offering novel insights into biophysical and biogeochemical processes. The NASA Carbon Airborne Flux Experiment (CARAFE) utilizes the NASA C-23 Sherpa aircraft with a suite of commercial and custom instrumentation to acquire fluxes of carbon dioxide, methane, sensible heat, and latent heat at high spatial resolution. Key components of the CARAFE payload are described, including the meteorological, greenhouse gas, water vapor, and surface imaging systems. Continuous wavelet transforms deliver spatially-resolved fluxes along aircraft flight tracks. Flux analysis methodology is discussed in depth, with special emphasis on quantification of uncertainties. Typical uncertainties in derived surface fluxes are 40—90% for a nominal resolution of 2 km or 16—35% when averaged over a full leg (typically 30 – 40 km). CARAFE has successfully flown two missions in the Eastern U.S. in 2016 and 2017, quantifying fluxes over forest, cropland, wetlands, and water. Preliminary results from these campaigns are presented to highlight the performance of this system.

## 1 Introduction

Accurate, quantitative, process-based understanding of current greenhouse gas (GHG) budgets (principally carbon dioxide ($CO_2$) and methane ($CH_4$)) is essential to projecting carbon-climate feedbacks and, hence, future climate (Friedlingstein et al., 2006; Schimel et al., 2015). While the atmospheric concentrations of these gases can be readily measured (Andrews et al., 2014), it is the sources and sinks at the Earth's surface and chemical conversion in the atmosphere that drive their changing abundances.

Global $CO_2$ budgets are typically constructed in a bottom-up sense from fossil fuel use inventories, estimates of ocean flux from solubility calculations, the measured time rate of change of the atmospheric $CO_2$ burden, perhaps a land use change/biomass burning emission term, and a land vegetation uptake flux inferred as the residual (Ciais et al., 2013; Le Quere et al., 2016). This construct provides very little information on the nature and distribution of the land flux or its

potential variations. Global $CH_4$ budgets are similarly under-constrained in detail (e.g., Bousquet et al., 2011; Dlugokencky et al., 2011; Worden et al., 2017).

Somewhat more specific information on source and sink distributions is commonly inferred from so-called top-down and bottom-up flux estimates. The former involves measuring atmospheric gradients in GHG mixing ratios and combining them with some estimate of volume transport to infer flux to/from the surface (flux inversion) (Chevallier et al.,

2010; Gurney et al., 2002). This method can be applied across a wide range of scales (global to 10's of km, depending on model resolution) but requires accurate transport characterization and intensive, high-accuracy GHG sampling. More recently, satellite measurements have been employed to provide increased sample density and coverage beyond that available from in situ measurements (Basu et al., 2013; Houweling et al., 2015). Bottom-up GHG flux estimates for terrestrial vegetation can be obtained from biogeophysical process models (Schaefer et al., 2008) and/or analysis of GHG

flux observations, the latter typically from large tower networks such as AmeriFlux (Boden et al., 2013; Jung et al., 2012). Top-down and bottom-up flux estimates often compare poorly and models disagree among themselves in regional flux estimation, with uncertainties typically exceeding 100% for continental-scale flux estimates (Hayes et al., 2012; Huntzinger et al., 2012). Validation of inferred fluxes is challenging due to both their coarse spatiotemporal resolution and a paucity of suitable observations. Observations of surface exchange at scales typically accessible to aircraft (1 – 100's of km) permit

validation of high-resolution emission inventories, probing of detailed biogeochemical interactions (e.g. drought stress), and general characterization of spatiotemporal gradients not resolvable from large-scale flux estimates. Several airborne methods have been developed for flux quantification, including mass balance approaches (Cambaliza et al., 2017; Karion et al., 2013a; Trousdell et al., 2016), inversions (Chang et al., 2014; Commane et al., 2017), and eddy covariance (Dabberdt et al., 1993; Desjardins et al., 1982). Each of these methods possess unique strengths and weaknesses in terms of their

measurement requirements, spatiotemporal resolution, and applicability to various processes (e.g. point source vs areal emissions, emission vs deposition/uptake). An exhaustive review is outside the scope of this work, which focuses on airborne eddy covariance.

Eddy covariance (EC) directly quantifies vertical turbulent fluxes in the atmospheric boundary layer. When measuring near the surface, fluctuations in vertical wind speed and scalar magnitude (e.g. temperature, gas concentration)

correlate positively/negatively if the surface is a net source/sink for that scalar. The time- or spatial-average product of vertical wind and scalar fluctuations (their covariance) thus yields a direct measurement of the flux at the measurement altitude. Extrapolation to the surface is possible with knowledge of the vertical flux divergence (change in flux with altitude), which is typically linear and can be obtained from flux measurements at multiple altitudes or independent constraints on the continuity equation (Conley et al., 2011; Lenschow et al., 1980). EC requires high-precision measurements of scalar and

vertical wind fluctuations throughout the range of turbulent timescales (up to several Hz for sampling mixed-layer eddies), which can be technically challenging. Uncertainties, dominated by the stochastic nature of turbulence, typically range from 20—80% for horizontal averaging scales of $1 - 30$ km but can exceed 100% when fluxes are small. Similar error ranges are reported for other surface exchange quantification methods (Cambaliza et al., 2014; Chang et al., 2014; Heimburger et al., 2017).

The main advantage offered by airborne EC is the ability to map gradients in surface exchange at relatively fine scales (~1 km) and over relatively broad regions (~100's of km). As with ground-based EC, airborne EC is not feasible over rough terrain (e.g. mountains), but moderate terrain (rolling hills) is acceptable (Misztal et al., 2014; Wolfe et al., 2015). The technique is especially well suited for disperse sources and sinks, such as vegetation and open water. Though not designed for single point sources, airborne EC can quantify aggregate fluxes over multiple small emitters, such as oil and shale gas production regions (Yuan et al., 2015). Airborne fluxes do not provide the long-term temporal information afforded by tower networks, but they can help characterize tower representativeness and/or extend tower observations to larger ecosystems (Chen et al., 1999; Kustas et al., 2006). In combination with spatially-resolved surface information (e.g. remotely-sensed vegetation properties), airborne fluxes can also help to refine surface exchange parameterizations (Anderson et al., 2008; Zulueta et al., 2013).

Airborne EC has elucidated surface-atmosphere exchange processes for more than three decades (Dabberdt et al., 1993; Desjardins et al., 1982; Lenschow et al., 1981; Ritter et al., 1992; Ritter et al., 1994; Ritter et al., 1990; Sellers et al., 1997). Recent GHG applications include evaluations of NEE over complex ecosystems (Miglietta et al., 2007; Zulueta et al., 2013) and quantification of $CH_4$ emissions from shale gas production regions (Yuan et al., 2015), agricultural areas (Desjardins et al., 2018; Hiller et al., 2014), and Arctic biomes (Sayres et al., 2017). The technique has also proven valuable for measurements of emissions, deposition and chemistry of reactive gases (Gu et al., 2017; Karl et al., 2009; Karl et al., 2013; Misztal et al., 2016; Misztal et al., 2014; Wolfe et al., 2015; Yu et al., 2017). Traditionally, airborne EC has been limited to small, low-flying aircraft (Gioli et al., 2004), but recent work has demonstrated successful flux observations from larger platforms (Wolfe et al., 2015; Yuan et al., 2015) that offer increased payloads for more complete atmosphere and ecosystem characterization..

The NASA Carbon Airborne Flux Experiment (CARAFE) is a new system engineered specifically for acquisition of airborne fluxes and related properties. To date, CARAFE has flown two 40-hour missions, one in September 2016 and another in May 2017. Table S1 in the supplementary information (SI) details the times and locations of each flight, and flight tracks are shown in Fig. 1f. Based out of NASA Goddard Space Flight Center's (GSFC) Wallops Flight Facility (WFF), flights targeted forest, farmland, wetlands and open water along the central U.S. East Coast. Here we describe the key components of the CARAFE payload and the methodology for deriving surface fluxes of $CO_2$, $CH_4$, sensible heat and latent heat. We utilize selected observations from both missions to demonstrate capabilities and performance. Future publications will present flux results for specific process representations in greater detail.

## 2 Platform and instrumentation

Here we describe the aircraft and core measurements included on both CARAFE deployments. Table 1 summarizes relevant specifications for each system. Key components include the aircraft, 3-D winds and associated meteorology, fast water vapor and greenhouse gas measurements, and a surface imaging system.

### 2.1 NASA C-23B Sherpa

The NASA C-23B Sherpa (Fig. 1a) is a high-wing, twin turboprop aircraft operated by NASA WFF. Modifications to support airborne science include addition of a variety of instrument ports and dedicated experimenter power (Fig. 1b—e). The Sherpa was first deployed during NASA's Carbon in Arctic Reservoirs Vulnerability Experiment (CARVE) to survey the abundance of $CO_2$, $CH_4$, and related gases in Canada and Alaska (Chang et al., 2014). Subsequent upgrades for CARAFE, as detailed below, now permit direct observations of surface exchange via eddy covariance.

The Sherpa is ideally suited for airborne flux measurements. Typical flight speeds of $80 \pm 10$ m s$^{-1}$ are sufficient for sampling turbulence statistics in the mixed layer. For example, a 10 Hz measurement corresponds to a data point every 8 m, and the scale of peak turbulence is roughly proportional to altitude (~100 m for a typical low-level leg). The nominal altitude envelope of 0.1—3 km facilitates both near-surface sampling and boundary layer profiling, and a range of ~1000 km (duration of 4—5 h) permits regional sampling from a single deployment location. With a payload weight and power capacity of 7000 lb and 6 kW, the Sherpa can support a full measurement suite for detailed in situ and surface observations. The payload described below uses roughly half of available weight and power.

Typical flux flight patterns consist of stacked level legs and vertical soundings. Level legs are mostly at an altitude of 90—150 m above ground level and range in length from 20—100 km. Occasional level legs higher in the boundary layer (200—400 m) provide a constraint for vertical flux divergence. Vertical soundings are required to assess boundary-layer depth and ideally occur in the middle of the target area at both the beginning and end of a sortie.

### 2.2 Meteorology and telemetry

The Sherpa Turbulent Air Motion and Meteorology System (TAMMS) is a suite of sensors for high-frequency measurement of horizontal and vertical wind vectors, pressure, and temperature. Winds are derived from five pitot static-pressure ports mounted on the radome (nose) of the aircraft (Brown et al., 1983; Thornhill et al., 2003). In essence, these sensors provide the velocity of air with respect to the aircraft. Combining this information with a high-quality GPS and Inertial Navigation System (Applanix 510) yields the velocity of air with respect to the Earth's surface. Calibration via standard aircraft maneuvers (Barrick et al., 1996) corrects for aircraft motion and specific features of the pressure field around the aircraft (Fig. S1). A hatch on the forward left side supports a Rosemount model 858 angle-of-attack probe for redundant vertical wind measurements and a Rosemount model 102 non-deiced total air temperature sensor housing coupled with a platinum sensing element (E102E4AL) (Stickney et al., 1990) for fast (~8 Hz) air temperature (Fig. 1c). A NASA Airborne Science

Data and Telemetry (NASDAT) system (https://asapdata.arc.nasa.gov/asf/sensors/nasdat.html) records data from these sensors at 20 Hz. The NASDAT also serves as a hub for GPS and network connections to other instruments.

The quality of the 3-D wind measurement hinges on the performance of the differential pressure measurements. For the 2016 campaign, both the radome system and the 858 probe were equipped with Honeywell PPT2 transducers. Afterward, spectral analysis of vertical wind speeds revealed anomalies at frequencies above 0.1 Hz. For the 2017 campaign the radome system was equipped with pressure sensors employing a higher sampling rate (Rosemount Model 1221), giving wind spectra more consistent with theoretical expectations (Fig. S2a). Comparison of the two wind systems for 2017 indicates that the 858 probe/Honeywell system under-samples $28 \pm 3\%$ of vertical wind variance, resulting in a systematic flux underestimate of ~24% (Fig. S2b). Division of all 2016 fluxes by a factor of 0.76 rectifies this bias in the mean, but additional random error arises from point-to-point variability (discussed further in Sect. 3.4). Due to this issue, fluxes and related quantities presented here will primarily utilize results from the 2017 mission.

An upward-looking photosynthetic photon flux density (PPFD) sensor (LI-COR LI-190R) is mounted on the wing and sampled at 1 Hz via the greenhouse gas system. This sensor is designed for level and stationary ground applications but performed well on initial flights. Section S1 in the SI describes post-processing of PPFD data to correct for aircraft attitude and sun position.

## 2.3 Fast water vapor

The NASA Langley Diode Laser Hygrometer, or DLH (Diskin et al., 2002), is an open-path infrared absorption spectrometer that uses a variation of wavelength modulation spectroscopy (Silver, 1992) to measure water vapor mole fraction. The DLH uses a laser locked to a water vapor absorption feature at ~1.395 μm and directs the beam from a transceiver mounted on the fuselage onto a retroreflector fixed to the upper surface of the Sherpa landing gear fairing (Fig. 1d). The returning light is collected and detected in the transceiver with a total roundtrip light path of ~2.5 m. Modulated signals are demodulated at twice the driving frequency (2F detection) and are converted to water vapor mole fraction using laboratory-determined laser characteristics, spectral parameters taken from the HITRAN 2012 database (Rothman et al., 2013), and the aircraft static pressure and temperature measurements. Raw data are processed at the instrument's native ~100 Hz acquisition rate. For CARAFE, data are averaged to 20 Hz with a typical precision (1σ) of 0.3% or better. Overall measurement accuracy is within 5%, based on field inter-comparisons on other airborne platforms (Jensen et al., 2017; Rollins et al., 2014).

## 2.4 Greenhouse Gas Suite (GHG)

The GHG system consists of several modified commercial analyzers coupled with custom hardware for fast gas flow and centralized data acquisition. Two Los Gatos Research (LGR) analyzers, one for $CO_2$ (model # 907-0020-1000) and the other for $CH_4$ and water vapor (model # 913-0014-0001), acquire mixing ratios at 10 Hz. The gas sampling system of each LGR is modified with a proportioning valve (IQ Valves, 0.234" orifice) coupled to a PID controller (Omega) to maintain a sample cell pressure of $140.0 \pm 0.1$ Torr throughout the Sherpa altitude range. Small pressure fluctuations in this range do not

noticeably impact measurement precision, and instrument pressure fluctuations are uncorrelated with wind speed fluctuations. Dry scroll pumps (Edwards nXDS15i) maintain a gas flow of ~35 SLM through each system. Laboratory evaluation of the LGR time response (Fig. S4) gives an e-fold flush time of 90 ± 16 ms, or an effective cutoff frequency (following the definition of Aubinet et al. (2016)) of 3.8 Hz. A Picarro G1301-m analyzer supplies an additional set of

$CO_2/CH_4/H_2O$ mixing ratios. Compared to the LGR analyzers, the Picarro provides a reduced duty cycle (~5 s for $CO_2$ and $CH_4$, ~15 s for $H_2O$) but greater precision and stability. Thus, the LGR systems provide fast measurements needed for EC flux calculations while the Picarro serves as an accuracy standard. For the 2016 deployment, a Picarro G2401-m replaced the G1301-m for the last four flights following a power supply failure on the latter. The specifications and accuracy are very similar for these two instruments, though the (newer) G2401-m offers a faster data rate (~0.5 Hz) and better precision (based

on in-flight comparison with the LGRs, Fig. 2d). Data streams from all analyzers, along with a GPS feed, are recorded via RS232 using a National Instruments CompactRIO controller.

The external gas inlet is a 6" length of 0.5" OD stainless steel tube (Fig. 1e). The tip of the inlet features a 15° rear-facing bevel to reject large particles. The inlet is mounted on an access hatch on the starboard side roughly 3 m aft of the nose. Directly behind the mounting plate, a tee connects to two identical lengths of Teflon PFA tubing (0.375" ID, 5.2 m

length). Each tube terminates at a high-flow inline Teflon particle filter (Entegris WGMXMBSS4) before connecting to one of the two LGR sample gas inputs. The Picarro analyzer sub-samples at 0.4 SLM from the LGR $CH_4/H_2O$ line. For typical flow rates and low-altitude flight, this configuration gives a gas sampling line residence time of 0.7 s and a Reynolds number of 4600.

Post-processing of GHG data occurs in several steps. First, data from all instruments are roughly time-aligned (to

within ~1 s for the Picarro and ~0.1 s for the LGRs) using both the internal timestamp of the data acquisition system and the GPS timestamp. Next, $CO_2$ and $CH_4$ observations are converted to dry mixing ratios using native $H_2O$ measurements (LGR $CO_2$ is corrected using $H_2O$ from the LGR $CH_4/H_2O$ sensor, time-aligned via lag-correlation). The corrections, which account for both density and spectroscopic effects, follow the quadratic form suggested for the Picarro G1301-m (Chen. et al., 2010; Rella, 2010) but with laboratory-derived, instrument-specific coefficients (Fig. S5). Picarro $CO_2$ and $CH_4$ mixing

ratios are calibration-corrected via small scaling factors (1.0026 for $CO_2$, 0.9994 for $CH_4$) based on comparisons to a NIST-traceable certified gas standard (NOAA ESRL) pre- and post-mission. Picarro G1301-m water vapor is calibration-corrected following the manufacturer's recommendation (Rella, 2010). Finally, for each flight, LGR observations of $CO_2$, $CH_4$ and $H_2O$ are linearly transformed to optimize agreement with the Picarro data. The transformation requires several operations, including 1) averaging dry mixing ratios to a common 1-Hz time-base, 2) smoothing LGR data to match the slower cell

throughput of the Picarro, 3) time-lagging the LGR (typically < 2 s) to optimize correlation with the Picarro, 4) calculation of fit coefficients for an ordinary least-squares fit (LGR = m*PIC + b), and 5) correction of LGR dry mixing ratios using the fit parameters. This procedure, akin to performing a flight-by-flight span and intercept calibration correction, rectifies calibration errors and flight-to-flight drift that may occur in the LGRs under different operating conditions. Figure 2 shows linear fits for all flights from 2016 (results for 2017 shown in SI Fig. S6). Linear correction factors vary little from flight-to-

flight. One exception to this procedure occurred on Flight 5 (16 Sept 2016), where a narrow concentration range gave an atypically poor water fit (Fig. 2d). Comparison of $H_2O$ observations between the LGR, Picarro, and DLH instruments (Sect. 4) corroborates this result. For this flight, the campaign-average slope and intercept were used to calibration-correct LGR $H_2O$. Campaign-average fit parameters were also used for two of the 2017 flights due to a failure in the Picarro system.

5   It is not currently feasible to calibrate the LGR systems in-flight due to high gas flow rates. Due to our correction procedure, we estimate that the accuracy limit of LGR $CO_2$ and $CH_4$ is degraded by a factor of 3 and 4, respectively, compared to the Picarro (Table 1). It may be possible to expand the above correction method to account for potential in-flight variability in LGR accuracy; however, as discussed later, measurement accuracy is a negligible contributor to total flux uncertainty for greenhouse gases.

**2.5 Surface Imaging System (SIS)**

The nadir viewing Surface Imaging Suite (SIS) consists of three cameras: a high resolution digital visible color camera (Nikon D7000), a thermal imager (FLIR A325sc), and a multiband camera specifically intended to observe vegetative health (MicaSense RedEdge). The Nikon D7000 is a single lens reflex camera incorporating a "DX" format 16.2 Megapixel CMOS imaging sensor (1.5X "crop factor") and 28 mm focal length lens. The lens was chosen to maximize field of view while

minimizing distortion. The shutter speed is generally set to 1/1000 second with an ASA/ISO of 640. Aperture is controlled by the camera, the manual focus is fixed at infinity, and images are captured on two SD memory cards. The FLIR A325sc is an uncooled microbolometer with sensitivity covering the range of 7.5 to 13.0 μm and a 320 x 240 imaging sensor. The lens focal length is 9.7 mm, providing a 45° x 34° field of view. Data is captured by small notebook computer. The MicaSense RedEdge is a five-band imager covering the spectral range of approximately 460 to 860 nm with blue, green, red, near-

infrared, and red-edge sensors. The camera has fixed focus and fixed field of view, and images are captured internally on an SC memory card. All three cameras are set for 1-s recording intervals, and time tagging and geolocation is done by remote GPS. Aircraft power (0.6 A) eliminates battery charging requirements, and the system is essentially autonomous. All imagery is downloaded after each flight and archived by the Wallops Remote Sensing Group. Figure 3 illustrates typical products acquired from this system.

25   The primary purpose of the SIS is to provide real-time qualitative information on surface characteristics and features that may influence gas and energy exchange. A rigorous comparison between imagery and airborne fluxes is non-trivial, as the flux footprint extends upwind of the aircraft with a typical half-width of several km, often exceeding the (altitude-dependent) swath width of the imagers. It is possible that an assumption of local surface homogeneity may be valid in some situations. Further work is needed to fully exploit the potential of combined surface imaging and in situ

observations.

## 3 Flux calculations

The flux methodology for CARAFE builds on previous work in airborne and ground-based EC. All calculations utilize a custom MATLAB toolbox, available upon request. Wavelet calculations utilize the framework described in Torrence and Compo (1998). The following discussion references both the time and spatial domains as appropriate, the two coordinates
being linked by leg-average aircraft speed.

### 3.1 Pre-processing

Data from the TAMMS, DLH, and GHG systems are averaged or re-gridded to a common 10 Hz time base. De-spiking is generally not necessary as data are quality controlled prior to archiving. Gas concentrations are provided as dry mixing ratios, eliminating the need for density corrections (Webb et al., 1980) to derived fluxes. Individual flux legs are identified
through inspection of heading, aircraft attitude, and vertical wind speed. We require that each leg be relatively level (±20 m altitude above ground). This altitude window is based on observed variability in the 2016 and 2017 data-sets and is not a hard limit. Data are also filtered to exclude aircraft roll exceeding 5 degrees to minimize potential artifacts in the vertical wind measurement. Wind vector rotation is not required as winds are already reported in a geodetic reference frame.

For each leg and each scalar time series, scalar data undergo mean removal and lag-correlation to the vertical wind
measurement. Lag times, with a typical range of 0—0.5 s, are determined through inspection of lag-covariance plots and held constant for each flight. Following Mauder et al. (2013), lag-covariance functions are calculated using fast Fourier transforms, and frequencies below 0.02 Hz (spatial scales greater than ~4 km) are removed to limit the influence of scalar trends on (co)variances. Note that these filtered lag-covariance functions are used for lag time determination and error calculations but not for calculation of actual fluxes.

Planetary boundary layer depth ($z_i$), required for calculation of flux errors and footprints, is assessed through examination of gradients in potential temperature and water vapor mixing ratio during vertical soundings. Due to mission constraints, soundings are not always located in the center of the target area or only occur at the beginning or end of a flight, making it difficult to quantify spatial or temporal variability in $z_i$. Thus, we assume a constant value for each target area. The uncertainty in this approximation is roughly ±100 m based on observed variability in BL depth for flights with multiple
soundings. Determination of exact BL depth is not critical, as calculated fluxes do not depend on BL depth and flux errors and footprints scale as a fractional power of $z_i$.

### 3.2 Ensemble average flux

Traditional ensemble average (EA) fluxes are calculated for each leg using vertical wind speed $w$ and scalar $s$ with the standard formulation.

$$F_{EA} = \langle w's' \rangle \tag{1}$$

Primes denote deviation from the mean. In addition, we derive spectra and cospectra using fast Fourier transforms (FFT) for each leg. These calculations are primarily for comparison with wavelet-derived quantities and are not used for scientific analysis, as terrain is often heterogeneous and observations are often non-stationary. A stationarity quality flag, $q_{stat}$, is calculated using the criteria of Foken and Wichura (1996):

$$q_{stat} = \left|1 - \overline{F_{EA,5}}/F_{EA}\right| \tag{2}$$

Here, $F_{EA,5}$ represents the mean of EA fluxes calculated on five evenly-sized sub-intervals within a leg. Ideally stationary legs will give $q_{stat} = 0$.

### 3.3 Continuous wavelet transform flux

The continuous wavelet transform (CWT) is a powerful and popular tool for time series analyses in atmospheric science. For a time series $x(t)$, wavelet coefficients $W_x(a,b)$ are calculated as a function of location (time or distance) and scale (frequency or wave number) by convolving the time series with a wavelet function ($\psi$).

$$W_s(a,b) = \int_{-\infty}^{\infty} s(t)\psi_{a,b}(t)dt \tag{3}$$

$$\psi_{a,b}(t) = \frac{1}{\sqrt{a}}\psi_0\left(\frac{t-b}{a}\right) \tag{4}$$

Here, the size and location of the wavelet are determined by the scale ($a$) and translation ($b$) parameters, respectively. Normalization of the wavelet function by $a^{-1/2}$ preserves the energy of the wavelet at different scales (Torrence and Compo, 1998). The CWT cospectrum is defined here as the cross-wavelet power of $w$ and $s$, $|W_w W_s^*|$, normalized by wavelet scale to correct for bias (Liu et al., 2007). We note that the latter operation is functionally equivalent to the common practice of multiplying FFT cospectra by frequency.

The wavelet function is actually a family of functions stemming from a "mother" wavelet, $\psi_0$. Mother wavelets are typically chosen based on the application, but a defining feature is localization in both the time and frequency (or distance and wave number) domains. This property, combined with scaling and translation, permits the wavelet transform to de-convolve contributions to time series variance along both the time (distance) and frequency (wave number) domains. In this work we utilize the Morlet wavelet, which is a plane wave modified by a Gaussian:

$$\psi_0(\eta) = \pi^{-1/4}e^{i\omega_0\eta}e^{-\eta^2/2}, \ \omega_0 = 6 \tag{5}$$

The Morlet wavelet is the standard choice for eddy covariance calculations, giving reasonable localization in both time and frequency domains (Schaller et al., 2017).

The CWT is well-suited for airborne fluxes, offering several advantages over traditional EA and FFT. Application of the CWT does not require stationarity, a condition that may be violated during long flight legs. This property also eliminates the need for signal detrending, improving quantification of long-wavelength flux contributions (Mauder et al., 2007). The technique provides a time series of fluxes along a flight track, removing the need to block-average homogenous sub-sections of the flight and giving relatively fine surface resolution that is essential when surveying patchy terrain. Further descriptions of CWT applications to airborne fluxes can be found elsewhere (Desjardins et al., 2018; Karl et al., 2009; Kaser

et al., 2015; Mauder et al., 2007; Metzger et al., 2013; Misztal et al., 2016; Misztal et al., 2014; Vaughan et al., 2016; Wolfe et al., 2015; Yuan et al., 2015).

Figures 4 and 5 illustrate a typical set of CWT flux results for $CO_2$ using observations over the Great Dismal Swamp, VA. The transect included sampling over forest, bog, and a small lake. The location of the lake at 15—22 km (distance relative to the start of the transect) is evident in the reduced variability of vertical wind speed (Fig. 4a). As expected, cospectral power (Fig. 4b) is mostly negative, enhanced over land (beginning and end of leg), and diminished over the lake. Integrating the local wavelet cospectrum over all scales yields the flux time series (Fig. 4c). In this example, the EA flux (also shown in Fig. 4c) is within 4% of the leg-average wavelet flux. This agreement can vary considerably from leg to leg, and in general we find that the agreement of EA and CWT fluxes is well correlated with stationarity (Fig. S7). Thus, comparison of EC and CWT fluxes may not be a useful quality metric for wavelet fluxes as suggested by some previous studies (Misztal et al., 2014). "Instantaneous" 10 Hz fluxes exhibit significant variability due to both instrument noise and the random nature of turbulence, and some averaging is required to obtain an estimate of "true" surface fluxes (discussed further below).

Averaging the local wavelet cospectrum along time/distance gives the global cospectrum (Fig. 5). Ogives (cumulative cospectra) are also shown. In this example, 90% of the cospectral power occurs at eddy scales of 2500—100 m, corresponding to sample frequencies of 0.03—0.7 Hz for an aircraft speed of 80 m s$^{-1}$. The comparable FFT cospectrum shows the same features but with more noise. The ogive also indicates that 99% of flux-carrying eddies occur at wavelengths longer than 54 m (frequencies lower than 1.4 Hz). This point serves as a reminder that fast instrument time response is less critical for sampling turbulence in the mixed layer as compared to the surface layer. Such measurement requirements scale with platform speed: on a faster-moving aircraft, an instrument must sample faster to resolve the same eddy scales.

### 3.3.1 Cone of influence

The cone of influence (COI) is the spectral region where wavelet coefficients may contain artifacts due to edge effects. As shown in Fig. 4b, the COI encompasses all scales at the ends of the time series and tapers toward larger scales (larger wavelengths/lower frequencies) near the center. Following the definitions of Torrence and Compo (1998) for the Morlet wavelet with $\omega_0 = 6$, the COI threshold scales linearly with distance from the beginning or end of the time series with a scaling factor of 0.73. This is an important consideration in flight planning. For example, if the largest flux-carrying eddies are ~5 km as in Fig. 5, the COI will fall below this value at distances greater than 5 km/0.73 = 6.8 km from leg edges. Thus, flight legs should be padded at least this distance on either end to ensure fully-resolved fluxes over a target area.

Treatment of the COI can impact scale-integrated CWT fluxes. Some previous studies exclude the COI before calculating fluxes (Vaughan et al., 2016), while others include it (Misztal et al., 2014). For short-length or higher-altitude flight legs, neglecting cospectral power within the COI may create systematic errors due to exclusion of larger-scale flux contributions. On the other hand, inclusion of covariance within the COI can lead to spurious fluxes, especially near the ends

of a leg and when fluxes are small. Here we develop a quality flag to quantify the potential impact of the COI and examine the effects of including or excluding the COI in scale-averaged fluxes.

The quality flag, $q_{COI}$, is calculated by interpolating global ogives (e.g. Fig. 5) onto the time series of COI threshold (given in wavelength or frequency units). Mathematically, for time $t$ and wavelength $\lambda$,

$$q_{COI}(t) = ogive_{w,s}(\lambda = COI(t)) \tag{6}$$

For this calculation, the ogive is calculated by integrating the absolute value of cospectral power to avoid negative values. Ranging from 0 to 1, this flag roughly represents the fraction of global cospectral power within the COI at each point in time. For example, for the data shown in Fig. 4b, the COI threshold at a distance of 5 km occurs at a wavelength of 3.7 km. The global ogive value at this wavelength is 0.12 (Fig. 5), indicating that 12% of the global cospectral power lies at lower frequencies, and thus $q_{coi} = 0.12$ for this time/location. The $q_{COI}$ flag can be used to filter scale-averaged CWT fluxes; for example, crosses in Fig. 4c indicate fluxes with $q_{COI} > 0.5$. Thus, more than half of the cospectral power resides within the COI for these times.

Figure 6 compares CWT and EA fluxes to quantify the potential impacts of COI treatment. This analysis is restricted to legs that are highly stationary ($q_{stat} < 0.1$) to ensure the quality of EA fluxes. Scale-averaged CWT fluxes are calculated by either including or excluding the cospectral power within the COI (see Fig. 4b). The resulting flux time series is subsequently filtered using a specified maximum $q_{coi}$ prior to leg-averaging. For example, a max $q_{coi}$ of 0.2 indicates that all points in the time series with more than 20% of cospectral power within the COI are excluded from the leg-averaged flux. CWT fluxes systematically under-predict EA fluxes by as much as 14% on average when omitting the COI and agree within 5% when including the COI. Thus, the extra systematic error from exclusion of the COI in the case without a $q_{coi}$ filter (rightmost points in Fig. 6) is 9%. Addition of the $q_{coi}$ filter reduces the CWT-EA discrepancy by removing CWT fluxes near leg edges where the COI influence is strongest. More restrictive filters (lower maximum $q_{coi}$) improve agreement at the expense of data density. For example, filtering for CWT fluxes with $q_{coi}$ less than 0.6, 0.4 and 0.2 removes 6%, 12% and 28% of 1 Hz CWT fluxes, respectively. We choose to include the COI when calculating CWT fluxes and assume fluxes are valid for $q_{coi} < 0.5$. From Fig. 6, this choice may reduce leg-average fluxes by ~2% on average.

### 3.3.2 Data gaps

Wavelet analysis is inherently designed for contiguous data, but data gaps are inevitable in field observations. Such gaps most commonly result from instrument calibrations or unfavorable aircraft attitude (e.g. evasive pitch or roll to avoid avian hazards). Since the wavelet algorithms of Torrence and Compo (1998) rely on fast Fourier transforms, these gaps must be removed or filled prior to performing the CWT. Several studies have suggested procedures for modifying wavelet basis functions to handle gaps (Frick et al., 1997; Frick et al., 1998; Mondal and Percival, 2008), but it is not clear how to implement such methods within our framework. SI Section S2 describes an empirical method that utilizes the covariance of scalar and vertical wind speed fluctuations to fill gaps with projected values. Even this method introduces some error in wavelet fluxes, especially in the immediate vicinity of a gap (Fig. S8). Transient errors are typically below 30% with a

spatial extent that scales with the width of the gap. To be conservative, we discard wavelet fluxes within a gap and on either side of a gap out to a distance equal to the gap width. In practice, gaps are rare and this procedure has a minimal impact on the total flux dataset.

### 3.4 Uncertainties

Uncertainty in EC fluxes arises from measurement limitations, sampling strategies, and the fundamental nature of turbulence. When extrapolating airborne fluxes to the surface, uncertainties in vertical flux divergence must also be considered. Methods to quantify flux errors are reviewed elsewhere (Langford et al., 2015; Mauder et al., 2013; Rannik et al., 2016). Here we leverage a combination of these methods and suggest a new technique to quantify the total random error in wavelet fluxes. Figure 7 summarizes individual error terms for the 2017 field campaign for all legs at altitudes below 200

m. Note that fluxes of $CH_4$ were negligibly small for most of the mission, thus the fractional errors are biased high. When flying over methane source regions (e.g. wetlands), $CH_4$ flux errors are comparable to those in $CO_2$ flux. In the following discussion, we adopt the convention of defining systematic errors as a fraction or percentage of the flux and random errors as an absolute value (in flux units). "Typical" error ranges quoted in the text refer to interquartile ranges (upper and lower box boundaries in Fig. 7) and are $1\sigma$.

### 3.4.1 Systematic errors

Under-sampling of turbulent fluctuations at both low and high frequencies creates systematic flux errors. For airborne observations, Lenschow et al. (1994) derive an upper limit for systematic error fraction due to under-sampling of low frequencies (long wavelengths) as a function of altitude above ground level ($z$), boundary layer depth ($z_i$), and leg length ($L$).

$$SE_{turb} \leq 2.2 \left(\frac{z}{z_i}\right)^{0.5} \frac{z_i}{L} \tag{7}$$

Typical $SE_{turb}$ ranges from 1.6% to 3.3% of observed fluxes (Fig. 7, light gray). The CWT utilizes data from the whole leg and thus theoretically captures all resolvable long-wavelength flux contributions at any given point in time. Thus, we assume this fractional error is constant for each point within a leg, irrespective of averaging length (e.g., a 1-km average is assigned the same $SE_{turb}$ as the whole leg).

Limited instrument time response is the main cause of high-frequency systematic errors for the CARAFE payload.
The open-path DLH instrument does not suffer from such limitations and thus serves as a time response standard. The ratios of cospectra for temperature, $CO_2$, $CH_4$ and LGR $H_2O$ fluxes against DLH $H_2O$ cospectra exhibit a characteristic decay at high frequencies that is well described by the transfer function (Horst, 1997)

$$H_s(f) = \frac{Co_{w,s}(f)}{Co_{w,DLH}(f)} = \frac{1}{1+(2\pi\tau_s f)^2} \tag{8}$$

Here, $f$ is natural frequency and $\tau_s$ is a characteristic response time for scalar $s$. Fitting Eq. (8) to global wavelet cospectra for all legs gives typical response times of 0.09 s for temperature and $CO_2$ and 0.1 s for $CH_4$ and LGR $H_2O$, consistent with lab tests (Fig. S4). Systematic error fraction due to response time is calculated for each leg as

$$SE_{RT} = \frac{\int_{-\infty}^{\infty} H_x(f) Co_{w,s}(f) df}{\int_{-\infty}^{\infty} Co_{w,s}(f) df} - 1 \tag{9}$$

Typical values for $SE_{RT}$ are $2 - 7\%$ of observed fluxes (Fig. 7, light green), assumed constant for each leg.

Accuracy in both the vertical wind and scalar measurements (Table 1) directly propagates into calculated fluxes. This systematic error ($SE_{acc}$), of unknown sign, adds uncertainties of 5% to $CO_2$ and $CH_4$ fluxes, 7% to sensible heat and DLH latent heat fluxes, and 8.6% to LGR latent heat flux (Fig. 7, dark green).

Systematic errors can be applied as a correction factor to fluxes (if of known sign) or be included as part of the total

uncertainty. Both practices are common among the airborne flux community (Gioli et al., 2004; Misztal et al., 2014). For the errors discussed above, $SE_{acc}$ is of unknown sign, while $SE_{turb}$ and $SE_{RT}$ should both increase the flux. We are, however, reluctant to employ the latter two as correction factors. $SE_{turb}$ represents an upper limit and thus may slightly "over-correct" the fluxes, while $SE_{RT}$ can become unrealistically large when fluxes are small due to the amplification of high-frequency noise by Eq. (9). Furthermore, systematic errors are typically small compared to random errors. Thus, we elect to include all

systematic errors in the total flux error and assume all error components are symmetric for simplicity. Total systematic error ($SE_{tot}$), given as a fraction of the flux over any interval, is then the root-sum-square of $SE_{turb}$, $SE_{rt}$, and $SE_{acc}$. Total systematic error is reported as a separate variable in flux archive files and may be used as part of the total error or as a correction factor (after removing the accuracy contribution) at the discretion of data end-users. One exception to this procedure is the treatment of systematic under-sampling of vertical wind variance in the 2016 observations (Sect. 2.2). For this particular

issue, all 2016 fluxes are corrected upwards by a factor of 1.32 (Fig. S2) and additional random error is incorporated as discussed below.

### 3.4.2 Random errors

Random flux errors arise from both uncorrelated noise in scalar measurements and the stochastic nature of turbulence. Errors due to uncorrelated instrument noise are calculated as

$$RE_{noise} = \sqrt{\sigma_{s,noise}^2 \sigma_w^2 / N} \tag{10}$$

Here, $\sigma_w^2$ is the variance of vertical wind speed over a whole leg and $N$ is the number of points. The noise variance in scalar $s$, $\sigma_{s,noise}^2$, is derived from the auto-covariance function as described elsewhere (Langford et al., 2015; Lenschow et al., 2000; Mauder et al., 2013). Relative to the total flux, this error is typically less than 5% for sensible and latent heat, 4—8% for $CO_2$ flux, and 11—96% for $CH_4$ flux (Fig. 7, indigo), with higher errors occurring when fluxes are small. For turbulence

sampling errors in airborne fluxes, Lenschow et al. (1994) suggest the following upper limit (as a fraction of total flux):

$$\frac{RE_{turb}}{F} \leq 1.75 \left(\frac{z}{z_i}\right)^{0.25} \left(\frac{z_i}{L}\right)^{0.5} \tag{11}$$

Values for $RE_{turb}$ typically range from 15% to 21% of leg-average flux (Fig. 7, dark gray). This equation suggests several strategies for reducing random errors, including reduced altitude, averaging over distance, or averaging over repeated legs.

The total random error is the square root of the sum of squared errors from Eq. (10) and (11). Alternatively, the total

error can be estimated empirically as the variance of the covariance between $s$ and $w$. (Finkelstein and Sims, 2001).

$$RE_{FS01} = \sqrt{\frac{1}{N}\sum_{p=-m}^{m}\left(\overline{s's'_p}\,\overline{w'w'_p} + \overline{s'w'_p}\,\overline{w's'_p}\right)} \tag{12}$$

Arguments $\overline{s'w'_p}$ and similar represent the unbiased cross-covariance or auto-covariance for lag $p$. Finkelstein and Sims (2001) suggest taking the summation over a sufficiently large $m$ to capture the integral time scale, somewhere in the range of 10—40 seconds. Mauder et al. (2013) further note that summing over too wide a range can give unrealistic results if a time

series contains trends. Following Mauder et al. (2013), we calculate the lagged cross- and auto-covariance functions using fast Fourier transforms and remove frequencies below 0.02 Hz  (spatial scales > 4 km) to limit trend effects. The maximum lag for the summation is set to 10 seconds based on comparison with the root-sum-square of $RE_{noise}$ and $RE_{turb}$, the latter representing a theoretical upper limit for total random error (Fig. S9a).

All of the above methods were originally developed for traditional EA eddy covariance, and it is not immediately

obvious how to extend error calculations to time-resolved CWT fluxes. Several previous studies (Karl et al., 2009; Misztal et al., 2014; Vaughan et al., 2016) have defined the random error for distance-averaged wavelet fluxes (1—10 km means) by substituting the averaging length for leg length ($L$) in Eq. (11). There are several issues with this approach. First, it does not account for errors due to instrument noise, which can comprise a significant fraction of the total error when turbulence-driven scalar variability approaches measurement precision. Second, Eq. (11) assumes that the flux calculated over distance

$L$ uses only observations acquired over that same window; however, the CWT inherently integrates information from the entire leg to derive (co)variances. In other words, the CWT flux for a 1 km region is not equivalent to the EA flux calculated using only wind and scalar observations sampled within that same 1 km.

We propose a novel method to calculate total random error along a wavelet flux time series. The technique is essentially an extension of Eq. (12), with lagged cross- and auto-covariance calculated using the scale and time-dependent

wavelet coefficients. In analogy to Eq. (24) of Torrence and Compo (1998), the covariance between variables $s$ and $w$ for time $t$ and lag $p$ is given by

$$\overline{s'w'_p}(t) = \frac{\delta j\delta t}{C_\delta}\sum_{j=j_1}^{j_2}\frac{|W_s(a_j,t)W_w^*(a_j,t_p)|}{a_j}\frac{N}{N-|p|} \tag{13}$$

Here, $\delta j$ is the wavelet spacing parameter (0.25 in our case), $\delta t$ is the sample interval (0.1 s for 10 Hz data), $C_\delta$ is a wavelet-specific reconstruction factor (0.776 for Morlet), and $a_j$ is the wavelet scale at index $j$. The last term on the right gives an

unbiased covariance estimate. When taken over multiple lags, Eq. (13) defines a wavelet cross-covariance function between $s$ and $w$ at each point in time (Fig. S10). Analogous equations apply for the other cross and auto-covariance terms.

Combining Eq. (12) and (13) thus gives the wavelet random error, $RE_{wave}$, as the variance of covariance along the time series. For this calculation the 1/N term in Eq. (12) is neglected, as the error is calculated for a single point in the time series and not the whole leg. As with the ensemble application of Eq. (12), the summation in Eq. (13) is restricted to scales with a corresponding Fourier frequency greater than 0.02 Hz (spatial scale < 4 km) and uses a lag range of ±10 s. This operation is computationally expensive for 10 Hz data, thus wavelet coefficients are averaged to 1 Hz before calculating $RE_{wave}$. This procedure yields comparable results to calculating errors from 10 Hz wavelet coefficients and then averaging to 1 Hz.

Two internal checks validate this approach. First, leg-averaged wavelet cross- and auto-covariance functions are virtually identical to those from ensemble-based lag calculations (Fig. S10). Second, leg-averaged $RE_{wave}$ values agree relatively well with both $RE_{FS01}$ and the root-sum-square of $RE_{turb}$ and $RE_{noise}$ (Fig. S9b-c). Typical $RE_{wave}$ values range from 10—50% of observed fluxes (Fig. 7, cyan).

As noted in Sect. 2.2, the 2016 dataset includes an additional random error component from damping of vertical wind variance. Using 2017 fluxes derived from the two wind datasets (Fig. S2), we estimate 1σ random errors in 2016 1Hz fluxes for sensible heat, latent heat, $CO_2$ and $CH_4$ of 50 W m$^{-2}$, 110 W m$^{-2}$, 7 µmol m$^{-2}$ s$^{-1}$, and 50 nmol m$^{-2}$ s$^{-1}$, respectively. This error is added directly to $RE_{wave}$ rather than in quadrature, as it is not unambiguously independent of $RE_{wave}$ (the latter being based on vertical wind and scalar (co)variance). The extra uncertainty does, however, reduce with averaging; note the tighter correlation for leg-average fluxes in Fig. S2b.

### 3.4.3 Vertical flux divergence

Extrapolation of airborne fluxes to the surface requires accounting for the change of flux with altitude, which should exhibit a near-linear profile in the boundary layer for non-reactive species. Vertical flux divergence can arise from several processes, as illustrated when considering the budget equation for a scalar $s$:

$$\frac{\partial F}{\partial z} = -\frac{\partial s}{\partial t} - \overline{U}\frac{\partial s}{\partial x} + Q \qquad (14)$$

Terms on the right-hand side respectively represent storage, horizontal advection (the product of horizontal wind speed and concentration gradient), and net in situ production or loss, which is negligible for long-lived GHGs. We neglect generally small contributions from vertical subsidence and horizontal turbulent fluctuations (Karl et al., 2013). It is possible to constrain each of the right-hand terms with a carefully-designed flight plan (Karl et al., 2013; Kawa and Pearson, 1989). Alternatively, flux measurements at multiple altitudes in the boundary layer provide a means of directly quantifying the flux divergence slope. Here we describe a procedure for deriving divergence corrections and discuss some of the associated challenges.

Calculation of the divergence correction begins with selection of a subset of flux observations. Figure 8 shows an example sensible heat flux profile for a series of 15 legs flown over the same 43 km forest swath (within a cross-track horizontal spread of ~2 km). Most legs on a typical CARAFE flight occur at low altitude (90—150 m), with only 1 to 3 legs at higher altitudes (200—400 m). Ideally the upper-level legs would be situated in the upper half of the boundary layer ($z/z_i$

> 0.5), but this is not always possible due to flight restrictions and the difficulty of determining boundary layer depth in real time. Fitting and error estimation requires a minimum of 3 legs and 2 altitudes. Chosen by visual inspection, these legs must be relatively close to one another in both space and time for a reliable fit. Diurnal variability is evident at low altitude over the 3 hour afternoon flight shown in Fig. 8, and in this case we limit the fit to legs with a solar zenith angle of less than 38°.

CWT fluxes are filtered prior to leg-averaging using both the COI quality flag ($q_{coi} < 0.5$) and a "proximity" filter. The latter requires that each point within a leg-average lies within 1 km of at least one point in every other leg. This proximity filter effectively trims the ends of each leg and limits the spatial average to regions of overlap. The choice of a 1 km radius is somewhat arbitrary and is a compromise between spatial overlap and data density, though we note that this is also a typical scale for a flux footprint. In some instances the proximity filter cannot be applied. For example, the fit in Fig. 8 uses legs

from both the "east" and "west" tracks. Though these tracks are spaced ~2 km apart, the forest is fairly homogeneous and the fit is more robust with the inclusion of more points.

Following data selection, divergence correction factors are calculated as follows. First, an error-weighted least-squares fit of the filtered and leg-averaged flux versus altitude gives the slope, $m$, and intercept, $b$. Next, a scaling factor $C_{div}$ is calculated as a function of altitude, with associated random uncertainty derived from fit parameter uncertainties ($\delta m$ , $\delta b$)

using standard error propagation.

$$C_{div}(z) = \frac{F(0)}{F(z)} = \frac{b}{mz+b} \tag{15}$$

$$RE_{div} = \left( \left( \frac{\partial C}{\partial m} \delta m \right)^2 + \left( \frac{\partial C}{\partial b} \delta b \right)^2 \right)^{0.5} = \frac{z}{(mz+b)^2} (\delta m^2 b^2 + \delta b^2 m^2)^{0.5} \tag{16}$$

Multiplication of the CWT fluxes by the associated $C_{div}$ then gives the surface-extrapolated flux. As shown in Fig. 7, the divergence scaling factor can vary considerably. "Typical" correction factors rescale fluxes sampled below 200 m by 10—

50%, and the additional random error from this correction is typically 5—30%. Divergence correction factors and their associated uncertainties are reported alongside CWT fluxes in archived data files.

The application of a "bulk" divergence correction inherently assumes that the correction is relatively invariant (within uncertainties) in both space and time for a given target region. It is possible to empirically test this assumption with observations. Figure S11a illustrates our tests for temporal and spatial variability in the divergence correction. Four of the

2017 flights included two high-level legs spaced 1.5—2.2 h apart, allowing derivation of divergence corrections at two different times for the same region. For this test we separately fit two legs (one high and one low) and calculate $C_{div}$ for each of the two sub-periods, then compare this to the $C_{div}$ value derived from all four legs combined. Fits are done on fluxes of $CO_2$, temperature, DLH $H_2O$, and LGR $H_2O$, giving a total of 16 scalar-flight pairs and 32 test cases. Based on this test, 95% of the sub-period $C_{div}$ values differ by less than 22% from the full-flight $C_{div}$ (Fig. S11b). To assess spatial variability, we

divide each set of legs into two sub-regions of equal length, calculate the divergence correction in each of the sub-regions, and compare this to $C_{div}$ calculated for the full region. Using the same set of fluxes as described above, this test includes 4 species, 4 flights, 2 divergence pairs per flight and 2 sub-regions per pair for a total of 64 test cases. Spatial variability is larger than temporal variability, with 95% of the sub-region $C_{div}$ values differing by less than 35% from the full region $C_{div}$

(Fig. S11b). In general, we find that the spatiotemporal variability of divergences corrections is within the calculated uncertainty for the divergence correction factor.

### 3.4.4 Total uncertainty

The total error for in situ fluxes is the root-sum-square of $SE_{tot}$ (multiplied by flux) and the empirical random error, $RE_{wave}$. When extrapolating to the surface, the divergence error is also added in fractional quadrature. Thus, the total fractional error for surface fluxes is

$$\frac{E_{surf}}{F_{surf}} = \left( SE_{tot}^2 + \left(\frac{RE_{wave}}{F}\right)^2 + \left(\frac{RE_{div}}{C_{div}}\right)^2 \right)^{1/2} \tag{17}$$

For $CO_2$, sensible heat and latent heat, typical $E_{surf}$ values range from 16—35% of the leg-averaged flux (Fig. 7, black). Values for $CH_4$ flux are significantly higher when considering the whole campaign since most of our sampling occurred in regions with scant methane emissions. Over the Alligator River and Dismal Swamp wetlands, where $CH_4$ emissions were significant, leg-average $E_{surf}$ values are $17 - 32\%$ of the leg-average $CH_4$ flux.

The above values are based on the 2017 dataset. For 2016 fluxes, consideration of the additional random error due to vertical wind variance under-sampling increases $E_{surf}$ by 1—2% of leg-averaged fluxes or 7—15% of 2 km average fluxes.

### 3.4.5 Error averaging

When averaging fluxes within a leg, fractional systematic errors are assumed constant while absolute random errors reduce as the mean of the root-sum-square of $RE_{wave}$ for each point in the average (see Eq. (1) in Langford et al. (2015)).

$$\overline{RE_{wave}} = \frac{1}{N}\sqrt{\sum RE_{wave}^2} \tag{18}$$

Divergence corrections ($C_{div}$) are averaged directly, while $RE_{div}$ is averaged as the root-mean-square (like Eq. (18) but without the extra factor of $N^{-1/2}$). Archived fluxes are reported at high resolution (1 Hz), but random errors due to turbulence sampling are large at these scales and some averaging is necessary to obtain statistically meaningful results. For example, the interquartile range in $CO_2$ surface flux uncertainty is 192—438% at the native 1 Hz (~80 m) resolution but improves to 40—90% when averaging to 2 km. These uncertainties are comparable to those reported in other airborne flux studies (e.g., Vaughan et al., 2016). Uncertainty reductions are also possible by averaging over repeated legs (Sect. 4.3).

### 3.5 Footprints

The flux footprint defines the distribution of surface sources/sinks contributing to the net flux observed at a given point. A simple 1-D metric for footprint size is the half-width, $dx_{0.5}$, defined as the distance along the mean horizontal wind that contains 50% of the surface flux (Karl et al., 2013; Weil et al., 1992).

$$dx_{0.5} = 0.9 \frac{\bar{U}z^{2/3}z_i^{1/3}}{w^*} \tag{19}$$

Here, w* is the convective velocity scale. Footprint scales from this equation typically range from hundreds of meters to ~10 km. For the CARAFE missions, estimates of $dx_{0.5}$ are reported alongside 1 Hz fluxes.

Robust comparison with ground observations or gridded model output may require an estimate of the full 2-D footprint. In this case, the "footprint" is effectively a spatial weighting function that can be applied to spatially-resolved quantities prior to integration and comparison with observed fluxes. For tower comparisons, fluxes can be filtered for times/locations where both platforms sampled the same footprint area. Applications of footprint models for airborne flux analysis vary in complexity. For example, Misztal et al. (2016) define a series of circles with radii equal to $dx_{0.5}$ and used these areas to integrate model-derived surface fluxes (with equal weighting within the circle). Sayres et al. (2017) utilize the 1-D parameterization of Kljun et al. (2004), while others have augmented the latter with a cross-wind distribution function (Metzger et al., 2012; Metzger et al., 2013; Vaughan et al., 2016). The recent 2-D parameterization of Kljun et al. (2015) (hereafter K15) is an attractive next step, both because it is based on the same Lagrangian framework as its 1-D predecessor and because the Matlab code is freely available. All required inputs for this parameterization are available from the CARAFE flux system, and footprints are theoretically calculable at any resolution up to the native 10 Hz resolution of the data stream. Equation (19) and K15 give comparable footprint half-widths for typical low-level legs below 200 m altitude (results not shown). Agreement is primarily a function of the stability parameter, $z/L_{ob}$ ($L_{ob}$ being the Obukhov length). For higher-altitude legs in strong convective conditions ($z/L_{ob} < -2$), footprint half-widths from K15 can be 2 to 4 times larger than $dx_{0.5}$; K15 note that this regime approaches the limits of applicability for the parameterization. In future work with the CARAFE dataset, we will evaluate what level of footprint complexity is required for comparison with surface fluxes.

## 4 Performance

Here we present a subset of results that illustrate the quality and performance characteristics of CARAFE observations. This evaluation is not exhaustive, and future studies will continue to assess the quality of fluxes through both internal quality controls and, when possible, comparison to other observations.

### 4.1 Spectral analysis

Figure 9 exemplifies quality metrics for a low-level leg with appreciable fluxes in all measured species. All cross-covariance functions (Fig. 9a) display strong peaks with similar integral time scales of ~3 s (defined as the time at the first zero crossing). Power spectra (Fig. 9b) for temperature, DLH-$H_2O$ and vertical wind speed measurements exhibit the $f^{2/3}$ power law ($f^{5/3}$ when not frequency-multiplied) in the inertial subrange, consistent with theory (Kaimal et al., 1972; Kaimal et al., 1976). In contrast, power spectra for $CO_2$, $CH_4$ and LGR-$H_2O$ show a shallower decay and exhibit a slope of ~1 above 0.4 Hz, indicative of white noise. The effects of instrument noise are also reflected in the increased variability in the cross-covariance functions at longer lag times and in the higher values of $RE_{noise}$ for these fluxes (Fig. 7). Despite the limited

precision of the closed-path analyzers at higher frequencies, cospectra with vertical wind generally agree for all scalars (Fig. 9c).

## 4.2 Water comparison

The CARAFE payload includes two independent water vapor measurements, providing a unique inter-comparison opportunity. The DLH system is open-path, fast-response, and field-proven through numerous prior missions. The LGR system is closed-path, displays reduced precision at turbulence-relevant frequencies, and had not flown prior to the 2016 mission. Figure 10 compares 1 Hz water mixing ratios and latent fluxes derived from both instruments. LGR water mixing ratios exhibit a small positive bias (slope = 1.05, $r^2$ = 0.995) throughout the mixing ratio range sampled during the 2017 CARAFE mission (0.3—1.5% by volume). This difference is well within the uncertainties of both instruments. LGR latent heat shows a somewhat larger bias relative to DLH latent heat (slope = 1.13, $r^2$ = 0.87). The source of this extra ~8% bias in fluxes is unclear; it may be related to the contact of sample air with surfaces (such as the 5.2 m sample line), though we would generally expect gas-surface interactions to dampen concentration fluctuations and thus reduce the flux. The bias is small compared to typical flux uncertainties, thus we will not explore the issue further here. Similar results were obtained for the 2016 mission.

## 4.3 Repeatability

The stochastic nature of turbulence imparts substantial random errors into small-scale flux measurements. For a typical 1 Hz (~75 m) flux, random errors are on the order of hundreds of percent. Averaging rapidly reduces this error; for example, random errors in a 2 km average flux are typically 30—50%. Precision is also improved by repeated sampling over the same swath of land, as long as changes in flux over the averaging period are small relative to random errors. Figure 11 shows $CO_2$ fluxes observed over five consecutive legs covering mixed farmland and forest. All legs were flown at an altitude of 100 m with a cross-track separation of less than 500 m. Typical $dx_{0.5}$ values are 800 ± 300 m, and leg-average fluxes vary by less than 12% with no discernable time trend. Random variability is evident in the 2 km average flux values. Nonetheless, all profiles exhibit the same general trend with lower fluxes (0 to -15 µmol m$^{-2}$ s$^{-1}$) at the beginning of the track, higher values (-10 to -30 µmol m$^{-2}$ s$^{-1}$) near the midpoint, and a sharp decline at the eastern edge (near the Atlantic coast and more urban areas). Averaging all legs together in each 2 km bin further reveals this trend. The 1σ random error for the multi-leg average is ~22% of the flux. Roughly 47% of the individual 2 km averages are contained within 1σ of the mean, and 72% within 2σ. This is somewhat less than the 68%/95% expected for a normal distribution; however, we do not necessarily expect a Gaussian distribution. Furthermore, some leg-to-leg variability is expected due to changes in wind speed and direction and thus the flux footprint. Overall, this result provides some additional confidence in our random error estimates.

# 5 Conclusions

The NASA CARAFE project aims to incorporate eddy covariance fluxes as a standard component of the airborne science toolbox. The C-23 Sherpa is well-suited for EC due to its particular balance of range, speed, and payload, though any aircraft that is amenable to fast 3-D wind measurements is a viable platform. The instrumentation deployed on the 2016 and 2017 missions provided observations of sufficient quality to calculate fluxes of sensible and latent heat, $CO_2$, and $CH_4$. Continuous wavelet transforms are key to unlocking the full potential of airborne fluxes, but only if utilized within a framework that properly accounts for all sources of uncertainty and the peculiarities of the technique (notably, the cone of influence and vertical flux divergence). Typical uncertainties in derived surface fluxes are 40—90% for a resolution of 2 km and 16—35% when averaged over an entire leg (typically 30—40 km). Initial results demonstrate sound spectral features of all measurements (with the exception of 2016 vertical winds), excellent agreement between closed and open-path water vapor observations, and reproducibility of horizontal flux gradients within random variability. Future efforts must continue to refine measurement and analysis techniques by both leveraging earlier work and acquiring new observations over a variety of conditions and surfaces. Inter-comparison with other methodologies for quantifying surface exchange, where possible, would also be valuable for both performance diagnosis and evaluation of multi-scale flux variability.

Direct observations of carbon and energy fluxes at regional scales offer unique opportunities for probing earth-atmosphere-biosphere interactions. This type of dataset is rare, and more work is needed to understand how such measurements can be applied to improve biophysical parameterizations and model or satellite-derived flux estimates. In particular, the spatiotemporal scales of airborne flux measurements – snapshot pictures in time over regional areas – are very different from the long-term, but spatially-sparse, tower flux observations typically available to the GHG community. Future efforts with the CARAFE dataset will include detailed comparisons to both ground observations and high-resolution earth system models, with the dual goals of developing techniques to upscale flux observations and furthering process-level understanding of biosphere carbon exchange.

Potential applications of airborne flux extend beyond the topics outlined here. Augmentation of the CARAFE payload with additional observations, such as mixing ratios and/or fluxes of carbonyl sulfide (Blonquist et al., 2011), may enhance the value of future datasets for diagnosing plant physiological responses. Co-located observations of surface properties such as solar-induced fluorescence and other markers of vegetation state/health could also prove synergistic for data interpretation. The methodology developed here is equally applicable to fluxes of reactive gases, including ozone, volatile organic compounds and oxidized nitrogen compounds. The process-level drivers of emission, deposition and transformation of these gases remain highly uncertain, and observational constraints on surface-atmosphere exchange are needed to challenge and improve air quality and chemistry-climate models. The combination of energy, carbon, and reactive gas fluxes may even provide new insights into the linkages between the biosphere, the atmosphere, and anthropogenic activities.

## 6 Data and code availability

All observations, 1 Hz fluxes and related quantities are publicly available through the CARAFE mission page at https://www.air.larc.nasa.gov/missions/carafe/index.html. Flux analysis code is available upon request from the corresponding author and will eventually be refined into a publicly-available Matlab toolbox.

## 5 Acknowledgements

We are deeply indebted to Piers Sellers, whose vision and boundless enthusiasm enabled this effort. We thank the management, pilots, crew, engineers, and mission support staff of WFF and the C-23 Sherpa for their selfless support and flexibility. We gratefully acknowledge the data archiving services provided by Gao Chen and Ali Aknan (NASA LaRC). We thank Ray Desjardins (Environment Canada) for helpful discussions, Sally Pusede and Laura Barry (U. VA) for their

assistance in operating the GHG instrument suite during the 2017 mission, and Dennis Gearhardt and Sean Kirby for their assistance with aircraft data acquisition. SIS team contributions from Ted Miles and Richard Mitchel are appreciated. Support for the CARAFE 2016 and 2017 missions was provided by GSFC Internal Research and Development, the NASA Carbon Monitoring System Program (NNH15ZDA001N-CMS), and the NASA HQ Earth Science Division. Wavelet software was provided by C. Torrence and G. Compo, and is available at http://atoc.colorado.edu/research/wavelets/. Finally,

we thank the two anonymous referees for their insightful and thorough comments.

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

**Table 1. Instrument specifications for the CARAFE payload.**

| Measurement | Instrument | Data Rate (Hz) | 1s Precision (@ 1Hz) | Accuracy |
|---|---|---|---|---|
| $CO_2$ | LGR | 10 | 330 ppb | 600 ppb[a] |
| | Picarro | 0.2 | 50 ppb | 200 ppb[b] |
| $CH_4$ | LGR | 10 | 2.0 ppb | 4 ppb[a] |
| | Picarro | 0.2 | 0.4 ppb | 1 ppb[b] |
| $H_2O$ | LGR | 10 | 200 ppm | 7%[c] |
| | Picarro | 0.07 | 100 ppm | 7%[c] |
| | DLH | 20 | 10 ppm | 5% |
| 3-D winds | TAMMS | 20 | 0.05 m s$^{-1}$ | 5% |
| Pressure | | | 0.003 mb | 5% |
| Vertical wind | Rosemount 858 | 20 | 0.05 m s$^{-1}$ | 5% |
| Temperature | Rosemount TAT | 20 | 0.05 K | 5% |
| Aircraft position | Applanix 510 | 20 | - | 100 m |
| Aircraft attitude | | | - | 0.005° |
| Telemetry | NASDAT | 20 | - | - |
| Visible imagery | Nikon 7000 | | | |
| IR imagery | FLIR 325sc | 1 | - | - |
| 4-band veg. health | MS RedEdge | | | |
| PPFD | LI-190R | 1 | - | 10% |

[a]Based on Picarro accuracy and variance of LGR-Picarro difference in 1 Hz observations.
[b]Based on laboratory calibrations and in-flight performance of similar instruments (Chen. et al., 2010; Karion et al., 2013b).
[c][b]Based on in-flight comparison with DLH.

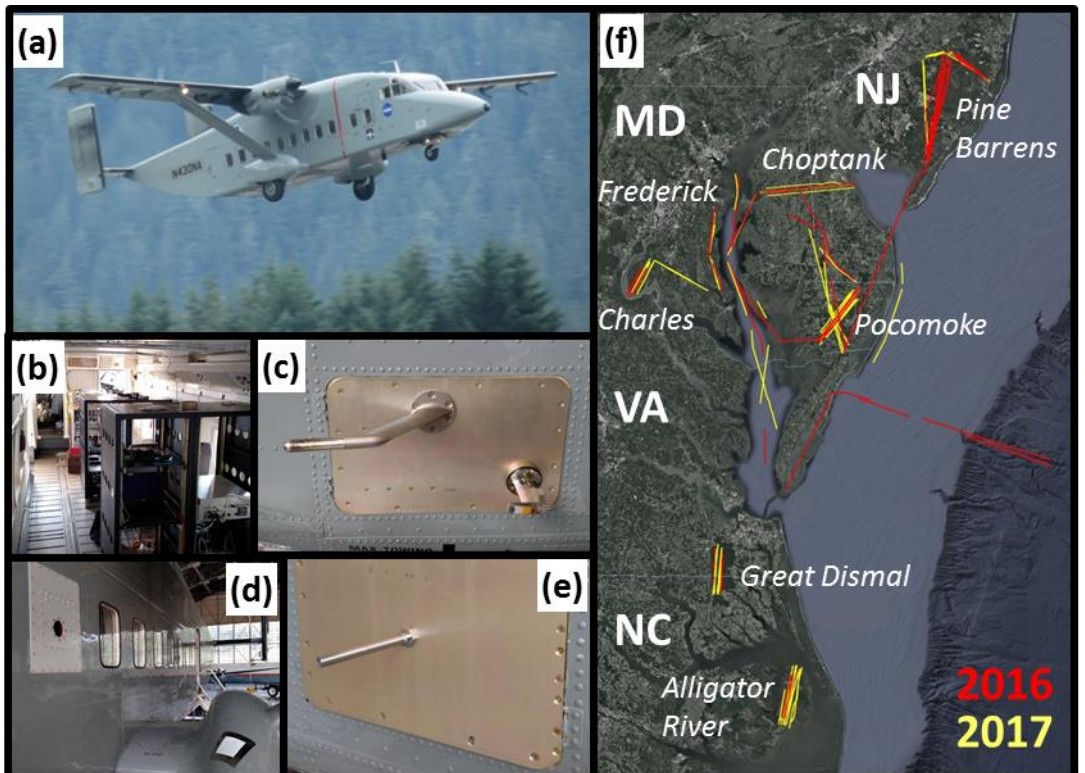

**Figure 1. The CARAFE payload. (a) The NASA C-23B Sherpa. (b) A view inside the cabin with all instruments installed. (c) The angle-of-attack (upper) and total air temperature (lower) probes. (d) The DLH window plate (upper left) and fairing-mounted target (lower right). (e) GHG inlet. (f) Flux leg flight tracks for both campaigns. Target locations indicated in italicized text correspond to those listed in SI Table S1.**

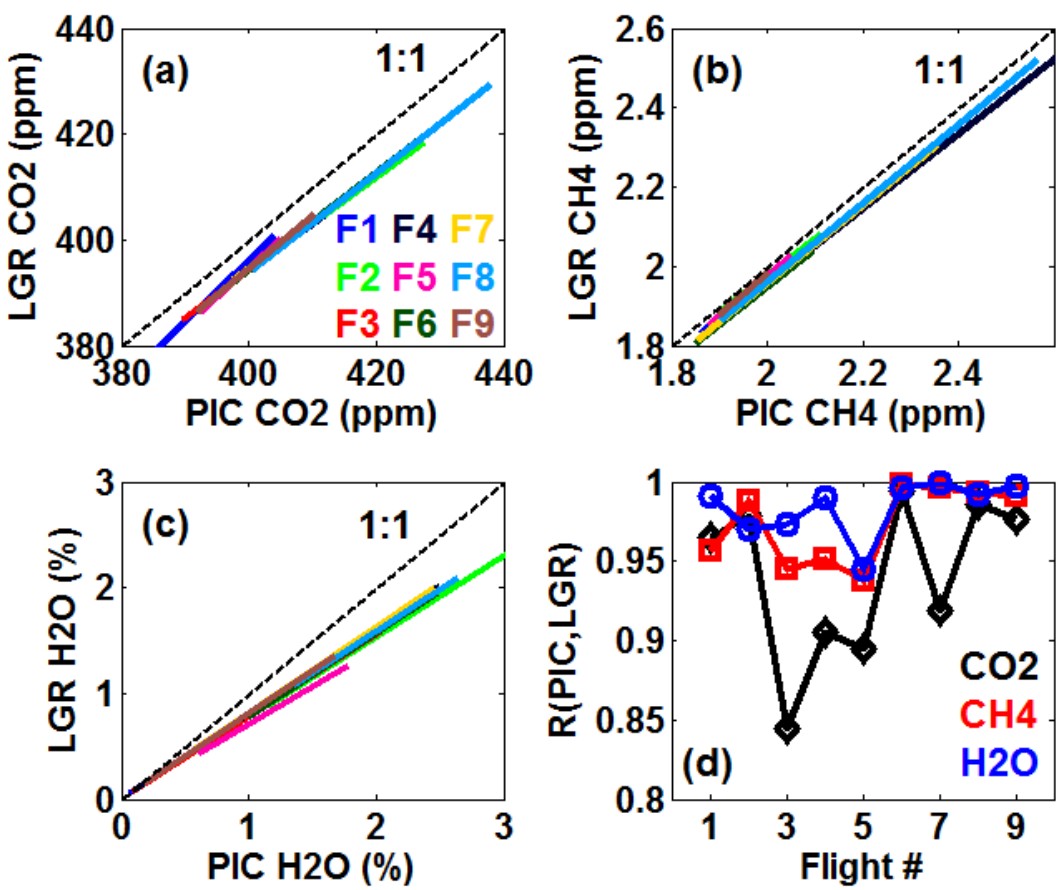

**Figure 2.** Least squares fit lines (a-c) and correlation coefficients (d) for Picarro and LGR dry mixing ratios of $CO_2$, $CH_4$ and $H_2O$ obtained during flights in 2016. Fits are colored by flight and shown only over the range of mixing ratios observed on each flight. Dashed lines in a), b) and c) denote a 1:1 correlation.

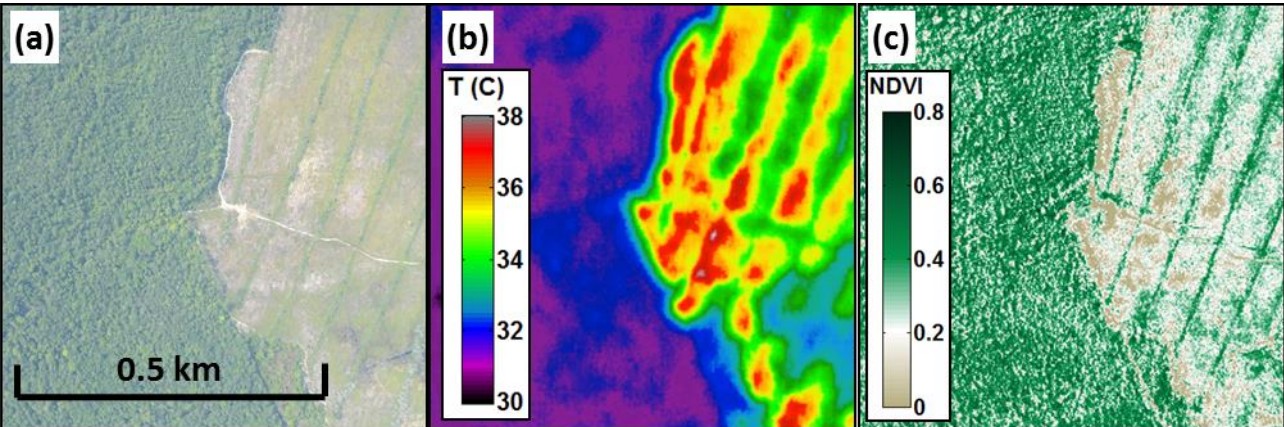

**Figure 3. Example imagery from the SIS recorded during the 09 September 2016 flight over Pocomoke forest. (a) Visible image from the Nikon 7000. (b) surface temperature from the FLIR A325sc. (c) Normalized difference vegetation index (NDVI) derived from the "red" (668 ± 10 nm) and "near IR" (840 ± 40 nm) bands of the MicaSense RedEdge sensor. Note, the color scale for the latter saturates at both ends.**

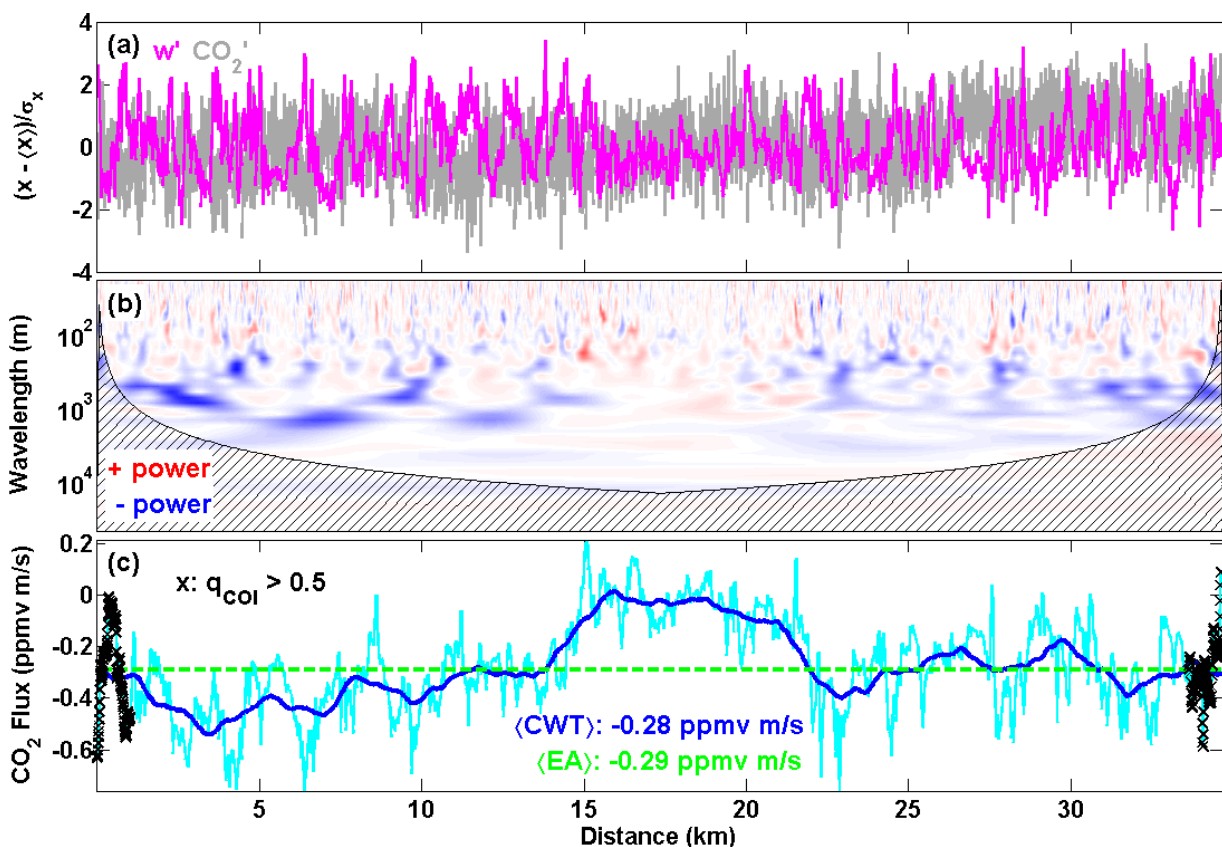

**Figure 4. Example wavelet CO₂ flux calculation from a flight leg at 130 m altitude over Great Dismal Swamp, VA on 16 May 2017.** (a) Normalized 10 Hz time series of vertical wind speed (w, magenta) and CO₂ (gray) fluctuations. (b) Local wavelet cospectrum. Red areas denote positive power, blue areas negative. Power is bias-corrected (multiplied by scale) as suggested by Liu et al. (2007). Hatched area indicates the cone of influence (COI). (c) Scale-integrated wavelet flux (cyan: 10 Hz, blue: 2 km running mean) and ensemble-average flux (green dash). Crosses denote points where 50% or more of the cospectral power lies within the COI.

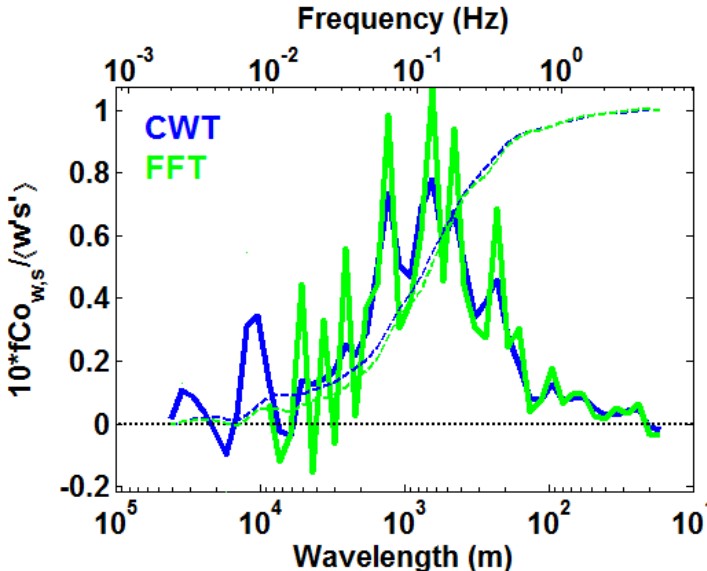

**Figure 5. Global cospectra of vertical wind and potential temperature for the leg described in Fig. 4, calculated by time-averaging the local wavelet cospectrum (blue) and fast Fourier transforms (green). Cospectra are scaled for display. Dashed lines show the cumulative integrals of the cospectra (ogives), normalized by total covariance.**

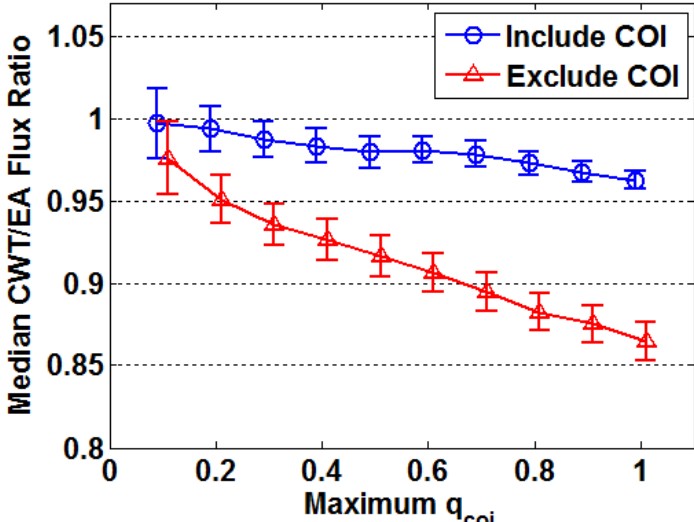

**Figure 6. Impact of treatment of the COI on wavelet fluxes. Wavelet fluxes are calculated either including (blue circles) or excluding (red triangles) cospectral power within the COI. In addition, wavelet time series are filtered using the $q_{coi}$ quality flag**
5  **(see text) at various thresholds prior to averaging over each leg. Symbols represent the median ratio of wavelet to ensemble-average fluxes for all scalars ($CO_2$, $CH_4$, H, LE-LGR and LE-DLH) and legs with stationarity flags of 0.1 or less. Error bars represent standard errors on the ratio.**

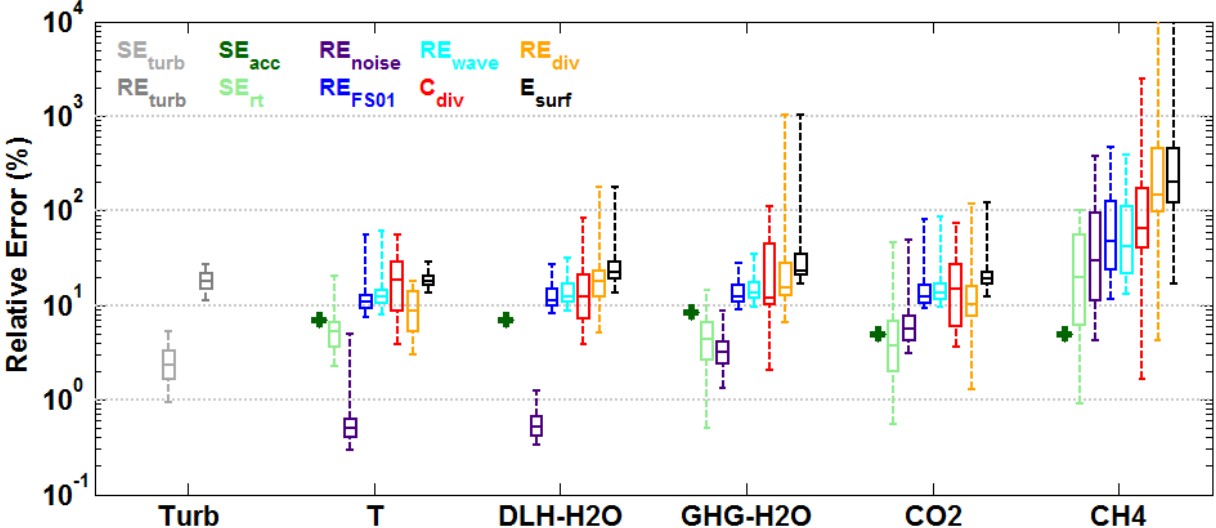

**Figure 7. Distribution of errors, normalized by leg-average fluxes, for all 2017 flight legs below 200 m altitude (97 legs total). In the box plots, the center line is the median value, box edges are 25th/75th percentiles and whiskers are 5th/95th percentiles. Turbulence sampling errors (systematic: light gray, random: dark gray) are the same for all scalars. Instrument-specific systematic error sources include measurement accuracy (dark green) and limited response time (light green). Random errors due to uncorrelated noise are shown in indigo. Empirical random errors, which inherently include both the turbulence and instrument noise components, are derived for both the leg ensemble (blue) and the wavelet time series (cyan). Also shown is the distribution of divergence corrections (red) and the additional fractional uncertainty in flux associated with this correction (orange). The total uncertainty in derived surface fluxes (black) includes the combined contributions from systematic, random, and divergence errors. Note that $SE_{RT}$ for DLH-H$_2$O is undefined as this instrument is the time response standard.**

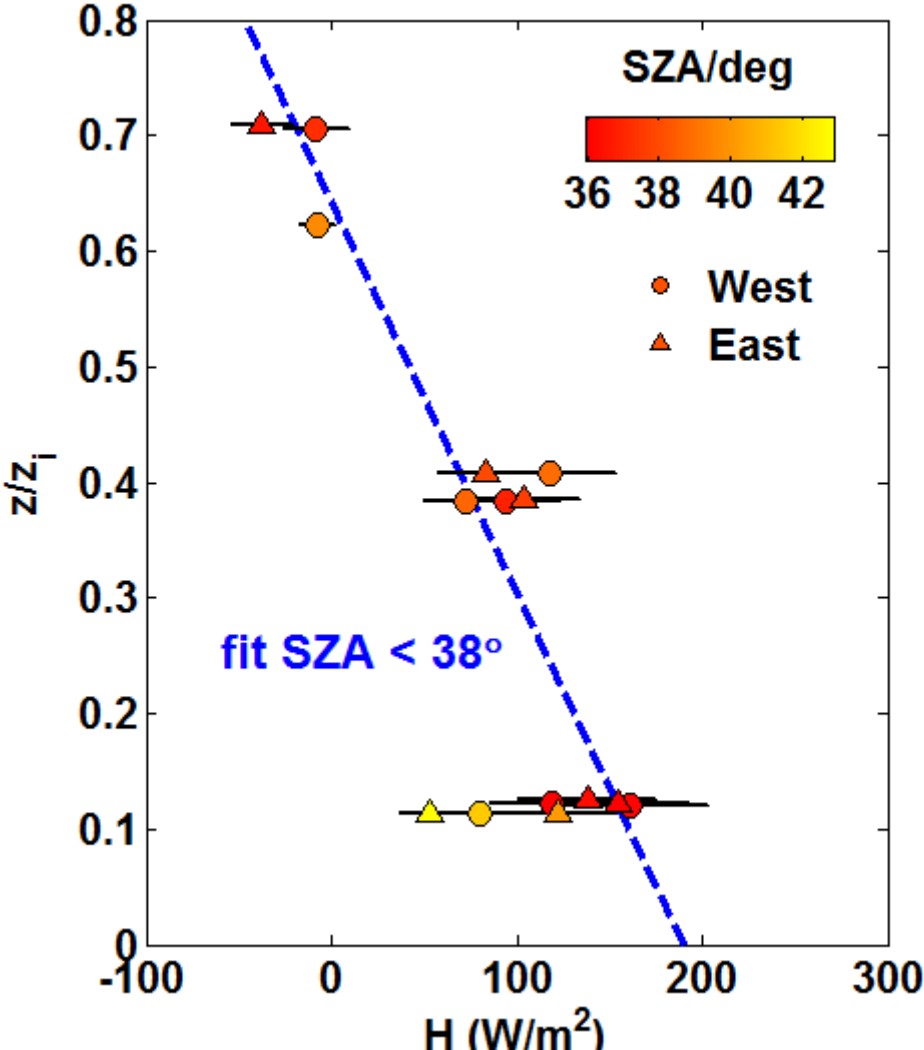

**Figure 8. Vertical profile of sensible heat flux observed over Pocomoke forest on 16 September 2016. Points represent mean CWT fluxes accumulated over ~43 km (9 min) of flight along two side-by-side tracks spaced ~2 km apart (circles: west tracks; triangles: east tracks). CWT fluxes with $q_{coi} > 0.5$ are excluded from the averages. Error bars represent total errors (systematic plus random). Data are colored by the mean solar zenith angle (SZA) during the leg. The blue dashed line is an error-weighted least-squares fit for all fluxes with SZA below 38°. Note that fluxes for this flight are corrected upwards by a factor of 1/0.76 for the vertical wind spectral artifact discussed in Sect. 2.2. The boundary layer depth for this flight was $z_i = 1070$ m.**

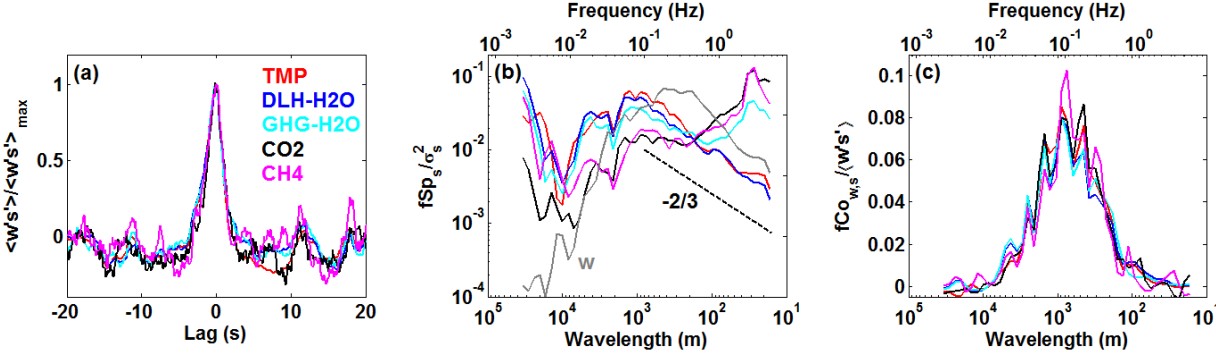

**Figure 9. Example vertical wind - scalar cross covariance functions (a), CWT power spectra (b) and CWT cospectra (c) for a 31 km leg at 100 m altitude ($z/z_i$ ~0.1) over Alligator River, NC on 26 May 2017. Cross covariance functions are normalized by peak covariance and time-shifted to align the peaks. Spectra are frequency-multiplied and (co)variance-normalized. The dashed line in (b) shows the expected -2/3 decay in the inertial subrange.**

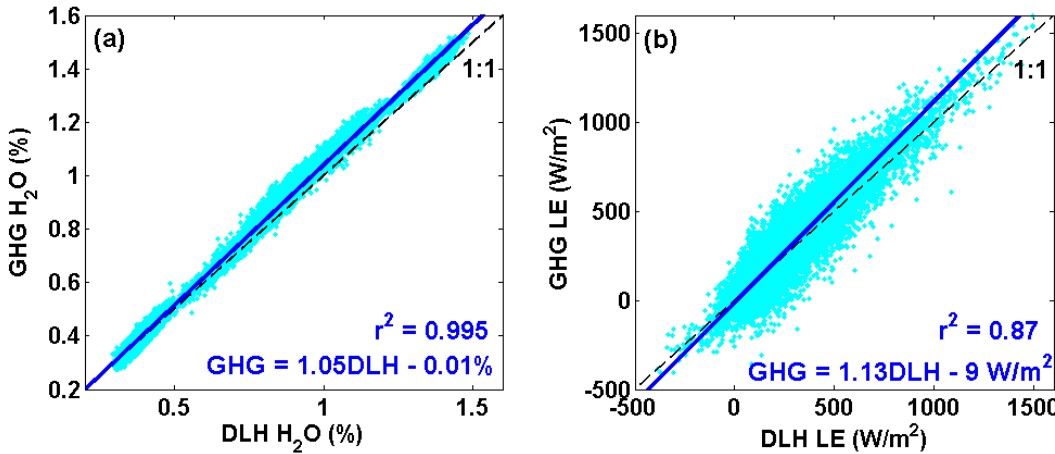

**Figure 10. Comparison of water vapor mixing ratios (a) and latent heat fluxes (b) for all legs of the 2017 mission. Cyan dots represent 1 Hz average data. Blue lines are reduced major axis linear fits.**

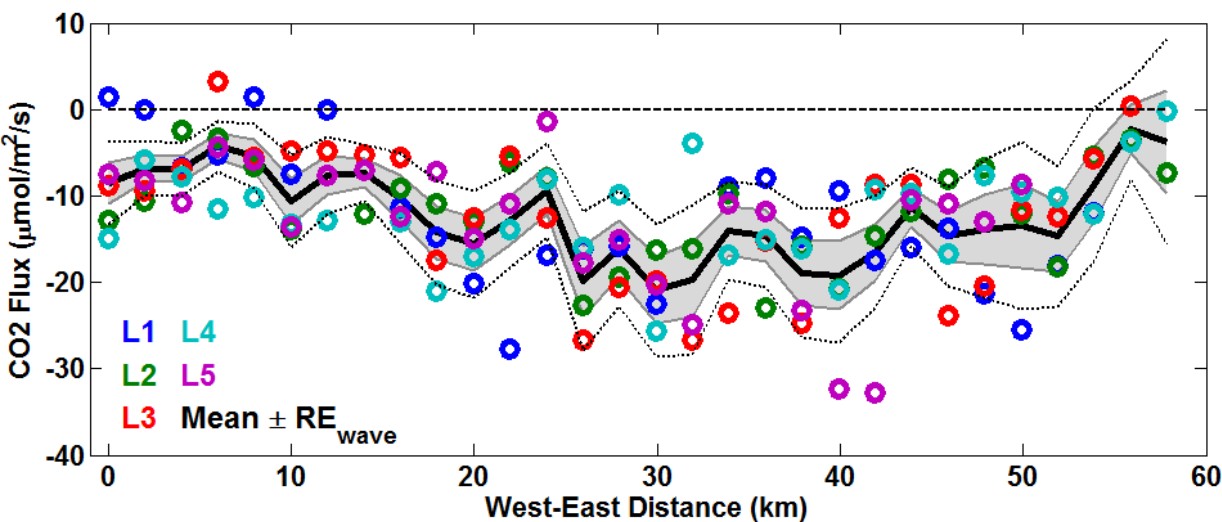

**Figure 11. Comparison of horizontal CO$_2$ flux profiles sampled over the same region of mixed cropland and forest on 04 May 2017. Colored circles represent 2 km average fluxes from each of five consecutive legs. The solid black line, shading, and dotted lines are the mean, 1σ random error, and 2σ random error, respectively.**