# Peer review of "The NASA Carbon Airborne Flux Experiment (CARAFE): Instrumentation and Methodology"

_Atmospheric Measurement Techniques, 2017_

## Referee Comment (RC1) · Anonymous Referee #1 · 12 Nov 2017

General Comment: The paper by Wolfe et al. describes airborne eddy covariance measurements on a C-23B Sherpa aircraft. It summarizes results from flights in the eastern US. The general topic is suitable for AMT, but there are a couple of issues that need to be addressed before publication. In particular, the discussion on errors needs revision.

Specific Comments:

It is not clear whether a Webb correction was necessary for CO2 and H2O fluxes, and how this was incorporated in the flux analysis code. The 10Hz humidity correction for the LGR instrument mentioned on page 5 (line 31) seems tricky – since there was a

redundancy of humidity measurements, a better experimental setup would have been to use a Nafion dryer for the EC system and just focus on CO2 and CH4 to avoid this problem all together. As the data are treated it is not clear to what extent the water vapor flux influences CH4 and CO2 fluxes, or how the correction procedure would degrade the precision of the flux calculation, given the large random errors of 10 Hz concentration datasets.

According to eq. 12 the turbulent random error should always be smaller than the combined error which includes instrument noise. Inspecting figure 7 actually shows the opposite for most tracers; the relative turbulent error is larger than REFS01 for T, H2O and CO2; this contradicts the theory. An explanation is needed – could there be a calculation error in the analysis code?

Eq. 12 is cast in the time domain. For aircraft measurements the time domain is not really meaningful. The discussion of errors should be handled in the spatial domain. For example, a cut off frequency of 0.02 Hz corresponds to a distance of 3.75 km at the aircraft speed of the C-23B Sherpa. The same criterion would correspond to a 12 km distance on a G5-aircraft.

The issue of spatial vs. temporal scale should be treated consistently throughout the manuscript. While the error discussion is treated in the time domain, some figures show a spatial, others a temporal scale. Figures 5 and 9 should be modified to show a spatial scale as well.

Total error: Systematic errors inherent to unresolved scales always lead to an under-estimation of fluxes and should be used to correct the data rather than adding these to a total error. Adding systematic errors to the total error is generally only admissible, if they are not separable from other errors or if their sign cannot be defined. Neither is true for SErt and SEturb. Additional systematic errors for surface fluxes arising from flux divergence are discussed separately but should probably be part of section 3.4.

Repeatability: it is mathematically not sound to simply average second moments as

presented in Figure 11 (see for example: https://www.eol.ucar.edu/content/combining-short-term-moments-longer-time-periods). Within the uncertainty of the presented data it might not make a large difference for Figure 11, but it would be worth double checking using the correct averaging formula.

Figure 6: the plotted differences are likely caused by a dramatic increase of systematic errors (eq. 7) towards the edges of the CWT – could the calculated flux ratios improve when accounting for these SE ? (e.g. by introducing a weighted SE along the CWT). To be more specific, the COI cuts off a substantial part of the frequency domain towards the edge of the CWT which should result in a systematic flux underestimation according to eq 7.

Minor Comments: Figure 8: How high was $z_i$? Figure 9a: A label for the $CO_2$ and $CH_4$ instrument should be added (e.g. LGR)

---

## Referee Comment (RC2) · Anonymous Referee #2 · 18 Jan 2018

Summary/General comments: Wolfe et al. describe the CARAFE aircraft, payload, and measurement methodology including flight data from campaigns in 2016 and 2017. Much of the manuscript focuses on the airborne eddy covariance method, how it is applied, and uncertainty analysis. The manuscript is well written and well placed in AMT. The authors have done a commendable job attempting to investigate the many challenges and sources of uncertainty in performing airborne eddy covariance. I do have some reservations and questions that need to be addressed. Once appropriate changes are made I would recommend publication.

Presentation/conceptual concern: The manuscript presents the CARAFE payload and

eddy covariance technique as a useful new tool for improving our understanding of carbon gas exchange. This tone underlies much of the manuscript, but the authors fall short of actually justifying, and this should be rectified. There is a cursory review of other airborne approaches that misses many techniques (such as the mass balance method), and the relative strengths/weaknesses are not really clearly highlighted. This isn't a problem if the manuscript focused on the CARAFE payload, but this would need to be addressed to assert the added value of airborne EC for CO2 & CH4. Even more so, the authors don't actually link observed eddy covariance to surface fluxes and provide added science value—it is made clear it is not known how to best link to horizontal spatial flux scales on the surface. More so, the error analysis suggests flux errors when considering the surface that can easily exceed 100%. I finished the manuscript wondering whether this approach was a wise usage of the aircraft and payload. Making high accuracy GHG airborne measurements from aircraft can be used with mass balance and different inversion systems to quantify fluxes with errors of ∼20%. With such an approach, larger areas can be covered with the aircraft as repeat legs are not needed and there are far less stringent requirements on level flight and surface characteristics, enabling the usage of far more data. Further, problems like the 2016 wind measurement error reported render all those flights of no scientific value because of the stringent requirements for EC. I commend the authors for their efforts and rigorous analysis, but at this point they cannot assert that airborne EC for CO2 & CH4 as presented in the manuscript provides added science value over more conventional accurate airborne sampling. I would actually think given the gaps in linking to horizontal surface domains, the tight restrictions on where the approach is useful, the limits imposed on flight area coverage, and the high fractional uncertainty, it is worth questioning if for surface carbon exchange this technique will add to addressing current science questions or whether accurate flight measurements for usage in inversions and mass balance approaches would be more scientifically fruitful. My suggestion is that the authors make changes in the abstract, introduction, and conclusion to more accurately capture this reality. The emphasis should be on the presentation of the

CARAFE payload. The extensive discussion of EC and uncertainty should remain, but a clear discussion of the limitations and that added science value is yet to be shown should be made clear.

Detailed comments:

Page 1 Line 16: not accurate as stated – exchange between surface and atmosphere only drives atmospheric abundance of some gases – not all atmospheric composition.

Page 1 Line 17: should modify to "potentially helping". Also, what are you defining as regional? Need spatial scale. Traditional airborne measurements can cover similar scales so would need to be specific and distinguish.

Page 1 Line 25-26: It does not follow from the paper that this system will further our understanding of ecosystem exchange – this has not been established.

Page 2 Line 3: the Dlugokencky reference is a very incomplete citation for such a broad statement.

Page 2 Line 4: The above described global approaches can also be defined as top-down and bottom-up. Need more specificity referencing spatiotemporal scales.

Page 2 Line 4-14: This is a very cursory coverage of other approaches that does not address many airborne approaches (mass balance, point source circling, eulerian/lagrangian inversion) that have been well established to evaluate fluxes at 10-100 km scales. NB those approaches are more flexible than EC and can deal with point sources that can be important for CO2 (power plants) and CH4 (lots of point sources). Addressing point sources is important for Carbon gases, and EC is ill-equipped for this. This point needs to be addressed.

Page 2, line 15: EC does not directly quantify surface-atmosphere exchange – it quantifies exchange between two atmospheric levels. An important distinction, as surface exchagne is inferred, which large errors induced due to flux divergence.

Page 2 line 24-26: As stated above, other airborne approaches are more flexible and have similar spatial capabilities.

Page 3 lines 1-2: This strong statement needs citation support.

Page 3 lines 28-31: This is illustrative of the very limited spatial domain that can be covered.

Section 2.2: I need to see more validation of winds. We should see the results from box patterns and other maneuvers done to test/validate winds, and thus be able to determine accuracy.

Section 2.2: The problem with the 2016 data is buried here. Based on this large, systematic problem, the authors decide not to use 2016 data. The authors should follow through and only show 2017 data (there is 2016 data still in many places). Further, this point should be made up front in the manuscript – small mistakes led to wind problems that rendered a whole deployment not useful for EC. This is illustrative of a major weakness to the CARAFE approach.

Page 5 line 4: Not sure where this comes from. I'd like to see more on this.

Section 2.4: I need to see more on the in-flight performance of the GHG analyzers. What is the accuracy in flight? Can the authors show the LGR analyzers show no vertical dependency compared with the Picarro?

Line 11: pressure fluctuations may impact accuracy however.

Page 6 line 5 & Figure 2: I am unclear on this linear transforming one instrument to the other. More clarity is needed. I also am concerned that this may not be appropriate for CO2 and CH4. When I look at figure 2 I get greatly concerned as the variation from flight to flight is actually very significant for gases that we care about fractions of a ppm (ppb for CH4). I also worry about inflight variations. We need more information on the validation of the GHG obs.

Section 2.5 (and figure 3): This section is somewhat of an aside. There is not other usage or discussion of this system.

Page 7 line8-9: This +-20 m requirement is very tight (as needed)- and will restrict the ability to use this data. Also, it should be made clear this is above ground and not asl. This makes it even harder to meet this requirement over terrain with any variability exceeding 20m. Further the 5 degree restriction is tough, but might it need to be tighter?

Page 7 Line 22: The problem is undersampling of the PBL depth can lead to systematic biases, and this is a major problem.

Page 11 Line 19: I'm confused, I had thought the authors earlier asserted 10Hz wasn't necessary, but here is seems this is a important error term.

Page 14 line 18: This is disappointing. If all these flight hours are being used there should be more planning for multiple altitude legs for this type of validation.

Page 15 lines 7-8: And these large uncertainties are a major problem for the approach. With relative uncertainties that push to 100% the utility of the technique is degraded.

Page 16 line 15-16: This is fair, but it means this manuscript has not established the utility of this approach.

Conclusions: I take issue with much of how the conclusion is written (see major comment above). This would be better served to summarize the aircraft system and payload, and then highlight the challenges in the EC approach and the resulting expected uncertainties.

Page 17 Lines 29-30: It has not been established that this approach should be a part of a standard toolbox.

Page 18 Lines 7-8: This isn't a new vector – as stated by authors the approach is old, and has been applied to Carbon before. The utility has never really been established,

particularly given the limitations, and that is why it has hardly been adopted.

Page 18 Line 17: This does not clearly follow.

Table 1: Methane and CO2 should be shown in ppb and ppm respectively (not fractional uncertainty).

Figure 1: The authors should indicate which flight legs are actually of use for EC on this plot – showing all the flight legs is misleading. 2016 data was deemed not useable, so should not be shown.

Figure 2: These are worryingly large to me. Also, this should show 2017 data as the authors don't use the 2016 flights.

Figure 7: This plot is sobering, and the log scale relative error brings into question the utility for CO2 and CH4.

---

## Author Comment (AC1) · 13 Feb 2018

Response to Anonymous Referee #1

We thank the Referee for their insightful comments. We have implemented a number of changes as outlined below.

**General Comment: The paper by Wolfe et al. describes airborne eddy covariance measurements on a C-23B Sherpa aircraft. It summarizes results from flights in the eastern US. The general topic is suitable for AMT, but there are a couple of issues that need to be addressed before publication. In particular, the discussion on errors needs revision.**

**Specific Comments:**

**It is not clear whether a Webb correction was necessary for CO2 and H2O fluxes, and how this was incorporated in the flux analysis code. The 10Hz humidity correction for the LGR instrument mentioned on page 5 (line 31) seems tricky – since there was a redundancy of humidity measurements, a better experimental setup would have been to use a Nafion dryer for the EC system and just focus on CO2 and CH4 to avoid this problem all together. As the data are treated it is not clear to what extent the water vapor flux influences CH4 and CO2 fluxes, or how the correction procedure would degrade the precision of the flux calculation, given the large random errors of 10 Hz concentration datasets.**

We do not perform a Webb correction, as the 10 Hz observations of CO2, CH4, and H2O (both LGR and DLH) are corrected to dry mixing ratios prior to calculation of fluxes. The interpolation of LGR H2O to the CO2 time base is not especially tricky, as 1) the gas sampling systems and native sampling rates are identical, and 2) the CO2-H2O correlation is sufficiently strong to provide good lag-correlation. We have added the following statement in the first paragraph of section 3.1:

*Raw gas concentrations are provided as dry mixing ratios, eliminating the need for density corrections (Webb et al., 1980) to derived fluxes.*

In our standard procedure, we calculate latent heat fluxes from DLH by first converting from mole fraction (moles per mole moist air, the native DLH measurement) to mixing ratio (moles per mole dry air), which theoretically negates the need for a Webb correction. As a test, we recalculated LE using mole fraction and applied a density correction following Eq. (23) of Webb et al. (1980). The full range of difference between the two methods is ±1%, and the normalized mean bias is 0.12%. The LGR instrument water corrections (Fig. S3) include both dilution and spectroscopic effects, thus it is somewhat more difficult to separate out the dilution component in our analysis code, but we expect a similar result. We further note that other researchers also avoid density corrections through similar approaches (Desjardins et al., 2018).

For the initial CARAFE deployments, we decided that not drying the GHG sample gas was preferable as the comparison of LGR H2O mixing ratios and fluxes against DLH provides a valuable performance

cross-check. Also, redundancy is insurance against instrument failure. We will consider drying the sample for future deployments.

**According to eq. 12 the turbulent random error should always be smaller than the combined error which includes instrument noise. Inspecting figure 7 actually shows the opposite for most tracers; the relative turbulent error is larger than REFS01 for T, H2O and CO2; this contradicts the theory. An explanation is needed – could there be a calculation error in the analysis code?**

The turbulence random error defined in Eq. (11) represents an upper limit (note the ≤ sign). Thus, we expect that the empirical total random errors, $RE_{FS01}$ and $RE_{wave}$, should be generally smaller than the root-sum-square of $RE_{turb}$ and $RE_{noise}$. Fig. S9 illustrates this point for *$RE_{wave}$*. We have added a plot to Fig. S9 to show a similar correlation for $RE_{FS01}$ and modified Sect. 3.4.2 as follows:

*The maximum lag for the summation is set to 10 seconds based on comparison with the root-sum-square of $RE_{noise}$ and $RE_{turb}$, the latter representing a theoretical upper limit for total random error (Fig. S9a).*

**Eq. 12 is cast in the time domain. For aircraft measurements the time domain is not really meaningful. The discussion of errors should be handled in the spatial domain. For example, a cut off frequency of 0.02 Hz corresponds to a distance of 3.75 km at the aircraft speed of the C-23B Sherpa. The same criterion would correspond to a 12 km distance on a G5-aircraft. The issue of spatial vs. temporal scale should be treated consistently throughout the manuscript. While the error discussion is treated in the time domain, some figures show a spatial, others a temporal scale. Figures 5 and 9 should be modified to show a spatial scale as well.**

We agree that it is more appropriate to cast discussion in the spatial domain when referring to turbulence scales or wavelet-derived fluxes. For certain aspects, however, the time domain is meaningful – specifically, with regards to instrumentation. For example, the characteristic response times described in Sect. 3.4.1 are inherent to each instrument and independent of platform speed. A similar argument holds for the influence of instrument noise on spectra shown in Fig. 9(b). With specific regard to Eqs. (12) and (13), we would obtain the same results (for this particular set of measurements) regardless of whether we use the temporal or spatial domain, as the Sherpa cruise speed is fairly constant for most flux legs (81 ± 9 m/s). The use of temporal domain is mostly a matter of convenience.

We have added the following statement to the beginning of Sect. 3:

*The following discussion references both the time and spatial domains as appropriate, the two coordinates being linked by leg-average aircraft speed.*

In addition, we have added/modified text throughout Sections 3 and 4 to better address spatial vs temporal scales (see esp. Sect. 3.3), and we have added spatial scales to Figs. 4, 5, and 9.

**Total error: Systematic errors inherent to unresolved scales always lead to an underestimation of fluxes and should be used to correct the data rather than adding these to a total error. Adding**

**systematic errors to the total error is generally only admissible, if they are not separable from other errors or if their sign cannot be defined. Neither is true for SErt and SEturb.**

Based on current literature, there seems to be no consensus in the flux community on how to handle systematic errors. Some groups lump systematic errors into total error as we have done (Misztal et al., 2014; Vaughan et al., 2016). Gioli et al. (2004) applies only a high-frequency correction, while Mauder et al. (2013) derives long-wavelength errors based on energy closure but advocates against using these to correct fluxes. Still other groups seem to ignore uncertainties entirely in their analysis (Desjardins et al., 2018; Sayres et al., 2017). Furthermore, $SE_{rt}$ is sometimes unreasonably large when spectra are noisy (P. 11, Line 29), and $SE_{turb}$ represents an upper limit. For these reasons, we believe it best to report SE as separate data columns and allow data users to decide how to treat these errors.

We have added the following discussion to the end of Sect. 3.4.1:

*Systematic errors can be applied as a correction factor to fluxes (if of known sign) or be included as part of the total uncertainty. Both practices are common among the airborne flux community (Gioli et al., 2004; Misztal et al., 2014). For the errors discussed above, $SE_{acc}$ is of unknown sign, while $SE_{turb}$ and $SE_{RT}$ should both increase the flux. We are, however, reluctant to employ the latter two as correction factors. $SE_{turb}$ represents an upper limit and thus may slightly "over-correct" the fluxes, while $SE_{RT}$ can become unrealistically large when fluxes are small due to the amplification of high-frequency noise by Eq. (9). Thus, we elect to include all systematic errors in the total flux error and assume all error components are symmetric for simplicity. Total systematic error ($SE_{tot}$), given as a fraction of the flux over any interval, is then the root-sum-square of $SE_{turb}$, $SE_{rt}$, and $SE_{acc}$. Total systematic error is reported as a separate variable in flux archive files and may be used as part of the total error or as a correction factor (after removing the accuracy contribution) at the discretion of data end-users.*

**Additional systematic errors for surface fluxes arising from flux divergence are discussed separately but should probably be part of section 3.4.**

We have moved this discussion to Sect. 3.4 and added sections on total error and error averaging.

**Repeatability: it is mathematically not sound to simply average second moments as presented in Figure 11 (see for example: https://www.eol.ucar.edu/content/combining-short-term-moments-longer-time-periods). Within the uncertainty of the presented data it might not make a large difference for Figure 11, but it would be worth double checking using the correct averaging formula.**

From the referenced link, the relevant formula here is

$$\overline{x'y'}^N = \frac{1}{N} \sum_{j=1}^{m} N_j (\overline{x'y'}^j + \overline{X}^j \overline{Y}^j) - \frac{1}{N} \sum_{j=1}^{m} N_j \overline{X}^j \frac{1}{N} \sum_{j=1}^{m} N_j \overline{Y}^j$$

Here, x and y correspond to scalar and vertical wind measurements and there are m sub-intervals of length $N_j$ included in the average over a total interval of length N. For any flux-relevant sub-interval,

however, the mean vertical wind should be sufficiently close to 0 that the terms on the right are negligibly small. Also, all $N_j$ are roughly equal in our averaging routines. In this case, the formula simplifies to an unweighted average, as used in this and many other studies.

We did check anyway, as the reviewer suggested. We find no appreciable difference in average fluxes using either averaging method.

**Figure 6: the plotted differences are likely caused by a dramatic increase of systematic errors (eq. 7) towards the edges of the CWT – could the calculated flux ratios improve when accounting for these SE ? (e.g. by introducing a weighted SE along the CWT). To be more specific, the COI cuts off a substantial part of the frequency domain towards the edge of the CWT which should result in a systematic flux underestimation according to eq 7.**

We assume that the reviewer is referring to the difference between the "exclude COI" and "include COI" cases. Exclusion of the COI from scale-averaging necessarily leads to a systematic underestimate of the true flux for the reasons the reviewer describes, and this is discussed in Sect. 3.3.1. Figure 6 provides an ensemble estimate for the resulting systematic error; in the case where we do not filter with the $q_{coi}$ flag (rightmost points), the error (taken as the difference between the "include" and "exclude" cases) is ~10% of the flux. We have added some text to this section to clarify this point.

It is difficult to develop a robust (time-dependent along the CWT) estimate of the systematic error resulting from the COI, as this area by definition represents a region where the wavelet transform suffers from limited information. In theory the $q_{coi}$ flag provides a rough means of doing this calculation: by first scale-averaging the CWT while excluding the COI, and then dividing by ($1-q_{coi}$) to correct for the fraction of cospectral power that was lost in the COI. The below plot shows the results of this calculation (black squares). The correction does indeed mitigate the systematic errors caused be excluding the COI for the ensemble of all fluxes. We hesitate to recommend such a correction for time-resolved CWT fluxes, however, as it inherently assumes that the globally-averaged ogive/cospectrum is representative of the local ogives/cospectra.

[Figure]

The purpose of the $q_{coi}$ flag is to allow filtering of fluxes that may suffer from large COI-related systematic errors. We believe this conservative strategy is preferable to attempting to recover cospectral power within the COI via a correction factor.

**Minor Comments: Figure 8: How high was zi?**

1070 m. We have added this info to the figure caption.

**Figure 9a: A label for the CO2 and CH4 instrument should be added (e.g. LGR)**

The "LGR" is meant to distinguish between the two water measurements. Since there is only one CO2 and CH4 flux measurement, we feel this change is unneeded. Actually the label should be GHG as this is the name of the system; we have modified Figs. 7, 9 and 10 accordingly.

References

Desjardins, R. L., Worth, D. E., Pattey, E., VanderZaag, A., Srinivasan, R., Mauder, M., Worthy, D., Sweeney, C., and Metzger, S.: The challenge of reconciling bottom-up agricultural methane emissions inventories with top-down measurements, Agr. Forest Met., 248, 48-59, 2018.

Gioli, B., Miglietta, F., De Martino, B., Hutjes, R. W. A., Dolman, H. A. J., Lindroth, A., Schumacher, M., Sanz, M. J., Manca, G., Peressotti, A., and Dumas, E. J.: Comparison between tower and aircraft-based eddy covariance fluxes in five European regions, Agr. Forest Met., 127, 1-16, 2004.

Mauder, M., Cuntz, M., Drue, C., Graf, A., Rebmann, C., Schmid, H., Schmidt, M., and Steinbrecher, R.: A strategy for quality and uncertainty assessment of long-term eddy-covariance measurements, Agr. Forest Met., 169, 122-135, 2013.

Misztal, P. K., Karl, T., Weber, R., Jonsson, H. H., Guenther, A. B., and Goldstein, A. H.: Airborne flux measurements of biogenic volatile organic compounds over California, Atmos. Chem. Phys., 14, 10631-10647, 2014.

Sayres, D. S., Dobosy, R., Healy, C., Dumas, E., Kochendorfer, J., Munster, J., Wilkerson, J., Baker, B., and Anderson, J. G.: Arctic regional methane fluxes by ecotope as derived using eddy covariance from a low-flying aircraft, Atmos. Chem. Phys., 17, 8619-8633, 2017.

Vaughan, A. R., Lee, J. D., Misztal, P. K., Metzger, S., Shaw, M. D., Lewis, A. C., Purvis, R. M., Carslaw, D. C., Goldstein, A. H., Hewitt, C. N., Davison, B., Beevers, S. D., and Karl, T. G.: Spatially resolved flux measurements of NOx from London suggest significantly higher emissions than predicted by inventories, Faraday Disc., 189, 455-472, 2016.

Webb, E. K., Pearman, G. I., and Leuning, R.: Correction of flux measurements for density effects due to heat and water vapour transfer, Quarterly Journal of the Royal Meteorological Society, 106, 85-100, 1980.

---

## Author Comment (AC2) · 13 Feb 2018

Response to Anonymous Referee #2

We thank the Referee for their insightful comments. We have implemented a number of changes as outlined below.

**Summary/General comments: Wolfe et al. describe the CARAFE aircraft, payload, and measurement methodology including flight data from campaigns in 2016 and 2017. Much of the manuscript focuses on the airborne eddy covariance method, how it is applied, and uncertainty analysis. The manuscript is well written and well placed in AMT. The authors have done a commendable job attempting to investigate the many challenges and sources of uncertainty in performing airborne eddy covariance. I do have some reservations and questions that need to be addressed. Once appropriate changes are made I would recommend publication.**

**Presentation/conceptual concern: The manuscript presents the CARAFE payload and eddy covariance technique as a useful new tool for improving our understanding of carbon gas exchange. This tone underlies much of the manuscript, but the authors fall short of actually justifying, and this should be rectified. There is a cursory review of other airborne approaches that misses many techniques (such as the mass balance method), and the relative strengths/weaknesses are not really clearly highlighted. This isn't a problem if the manuscript focused on the CARAFE payload, but this would need to be addressed to assert the added value of airborne EC for CO2 & CH4. Even more so, the authors don't actually link observed eddy covariance to surface fluxes and provide added science value; it is made clear it is not known how to best link to horizontal spatial flux scales on the surface. More so, the error analysis suggests flux errors when considering the surface that can easily exceed 100%. I finished the manuscript wondering whether this approach was a wise usage of the aircraft and payload. Making high accuracy GHG airborne measurements from aircraft can be used with mass balance and different inversion systems to quantify fluxes with errors of 20%. With such an approach, larger areas can be covered with the aircraft as repeat legs are not needed and there are far less stringent requirements on level flight and surface characteristics, enabling the usage of far more data. Further, problems like the 2016 wind measurement error reported render all those flights of no scientific value because of the stringent requirements for EC. I commend the authors for their efforts and rigorous analysis, but at this point they cannot assert that airborne EC for CO2 & CH4 as presented in the manuscript provides added science value over more conventional accurate airborne sampling. I would actually think given the gaps in linking to horizontal surface domains, the tight restrictions on where the approach is useful, the limits imposed on flight area coverage, and the high fractional uncertainty, it is worth questioning if for surface carbon exchange this technique will add to addressing current science questions or whether accurate flight measurements for usage in inversions and mass balance approaches would be more scientifically fruitful. My suggestion is that the authors make changes in the abstract, introduction, and conclusion to more accurately capture this reality. The emphasis should be on the presentation of the CARAFE payload. The extensive discussion of EC and uncertainty should remain, but a clear discussion of the limitations and that added science value is yet to be shown should be made clear.**

Most of the above comments are handled more specifically below. We have substantially revised the introduction and error discussion following the comments of both reviewers.

Regarding the utility of airborne EC, the wealth of work cited in the introduction is, we hope, sufficient evidence. We do not claim that airborne EC is a replacement for other methods. All methods have unique strengths and weaknesses, and all are needed for scientific progress on our understanding of surface exchange, which is a very broad umbrella.

**Detailed comments:**

**Page 1 Line 16: not accurate as stated – exchange between surface and atmosphere only drives atmospheric abundance of some gases – not all atmospheric composition.**

We have changed "drives" to "strongly influences."

**Page 1 Line 17: should modify to "potentially helping". Also, what are you defining as regional? Need spatial scale. Traditional airborne measurements can cover similar scales so would need to be specific and distinguish.**

Wording changed. We define local to regional scales as 1 – 1000 km.

**Page 1 Line 25-26: It does not follow from the paper that this system will further our understanding of ecosystem exchange – this has not been established.**

We have changed the last sentence as follows:

*Results from these campaigns highlight the performance of this system and its potential to provide fresh observational constraints on ecosystem exchange.*

**Page 2 Line 3: the Dlugokencky reference is a very incomplete citation for such a broad statement.**

We have added several additional references.

**Page 2 Line 4: The above described global approaches can also be defined as topdown and bottom-up. Need more specificity referencing spatiotemporal scales.**

We have changed the wording to clarify here. The first paragraph refers to abundance-based budgets, and the second to flux distributions/budgets.

**Page 2 Line 4-14: This is a very cursory coverage of other approaches that does not address many airborne approaches (mass balance, point source circling, eulerian/lagrangian inversion) that have been well established to evaluate fluxes at 10-100 km scales. NB those approaches are more flexible than EC and can deal with point sources that can be important for CO2 (power plants) and CH4 (lots of point sources). Addressing point sources is important for Carbon gases, and EC is ill-equipped for this. This point needs to be addressed.**

We have added several paragraphs to the intro mentioning these other methods and discussing the strengths/weaknesses of airborne EC.

**Page 2, line 15: EC does not directly quantify surface-atmosphere exchange – it quantifies exchange between two atmospheric levels. An important distinction, as surface exchagne is inferred, which large errors induced due to flux divergence.**

We have changed the wording as follows:

*Eddy covariance (EC) directly quantifies vertical turbulent fluxes in the atmospheric boundary layer.*

**Page 2 line 24-26: As stated above, other airborne approaches are more flexible and have similar spatial capabilities.**

This paragraph focuses on the distinction between ground and airborne EC, but has been deleted in revision.

**Page 3 lines 1-2: This strong statement needs citation support.**

We have deleted this sentence as part of the intro revisions. This is an inherent feature of the wavelet transform due to its localized nature.

**Page 3 lines 28-31: This is illustrative of the very limited spatial domain that can be covered.**

The total 2017 dataset includes over 3000 km of surface fluxes. We would not call this limited.

**Section 2.2: I need to see more validation of winds. We should see the results from box patterns and other maneuvers done to test/validate winds, and thus be able to determine accuracy.**

We have added a new figure for wind calibrations (Fig. S1) and reference to this figure in the text.

**Section 2.2: The problem with the 2016 data is buried here. Based on this large, systematic problem, the authors decide not to use 2016 data. The authors should follow through and only show 2017 data (there is 2016 data still in many places). Further, this point should be made up front in the manuscript – small mistakes led to wind problems that rendered a whole deployment not useful for EC. This is illustrative of a major weakness to the CARAFE approach.**

We call out this problem explicitly in the text and provide a figure in the SI to illustrate the effects. The upshot is an unfortunate but well-defined systematic error that is partially correctable. While it reduces the utility of the 2016 data, it does not render the whole deployment useless. We have added the following text to this section to better clarify/quantify:

*Comparison of the two wind systems for 2017 indicates that the 858 probe/Honeywell system misses 28 ± 3% of vertical wind variance, resulting in a systematic flux underestimate of ~24% (Fig. S2b). Division of all 2016 fluxes by a factor of 0.76 rectifies this bias in the mean sense, but additional random error arises from point-to-point variability. Based on differences in 2017 fluxes derived from*

*the two wind datasets, we estimate 1σ random errors in 2016 1Hz fluxes of sensible heat, latent heat, $CO_2$ and $CH_4$ of 50 W m$^{-2}$, 110 W m$^{-2}$, 7 μmol m$^{-2}$ s$^{-1}$, and 50 nmol m$^{-2}$ s$^{-1}$, respectively. These errors are in addition to those discussed in Sect. 3 and reduce with averaging (note the tighter correlation for leg-average fluxes in Fig. S2b).*

We have also added several notes about this in Sect. 3.4, and the archive data files will be updated accordingly.

The 2016 flux data appears only one time, in Fig. 8. We prefer to leave this as is; it is our best example of the effects of diel cycles on divergence.

We have added a sentence in the intro stating that the measurement requirements make this a challenging technique, but we might also note here that such problems are not isolated to airborne EC. For example, a recent mass balance analysis on CH4 emissions was retracted following discovery of an issue in another aircraft wind system (Ren et al., 2018).

**Page 5 line 4: Not sure where this comes from. I'd like to see more on this.**

We have added several references to support DLH accuracy. Also, we note the excellent agreement with independently-calibrated GHG water vapor shown in Fig. 10.

**Section 2.4: I need to see more on the in-flight performance of the GHG analyzers. What is the accuracy in flight? Can the authors show the LGR analyzers show no vertical dependency compared with the Picarro?**

We do not currently calibrate the GHG system in-flight. Chen. et al. (2010) have demonstrated the stability of the Picarro G1301-m under flight conditions and suggest that ground-only calibrations are sufficient to obtain an accuracy of 0.05 ppm for $CO_2$. Karion et al. (2013) suggest a total uncertainty (including precision and other error terms) of 0.23 ppm for CO2 and 2.3 ppb for CH4 from another airborne G1301-m system operating over a broader range of altitudes than those covered by CARAFE. Based on these values and considering that we do not calibrate in-flight, we have revised our accuracy estimates in Table 1 to be more conservative.

The LGR is necessarily less accurate than the Picarro because of our fitting procedure, by roughly a factor of 3-4 based on comparisons between the final 1 Hz data products. This information is included in our data files but was accidentally omitted in Table 1, and we have added it now.

High accuracy is not critical for airborne EC, as evidenced by our error discussion; accuracy is generally a small term in the total flux error budget (Fig. 7). Indeed, in the case of CO2 and CH4, the accuracy error is dominated by the 5% winds uncertainty.

As shown in the plots below (colored lines denote different flights), there is some vertical dependence in the difference between the LGR and Picarro for CO2 measurements in 2016, the reasons for which are unclear at this time (possibly improved pressure control). The bulk of surface flux observations

come from measurements below 500 m, where the difference is generally within our stated uncertainty. This issue has little bearing on derived fluxes.

[Figure]

**Line 11: pressure fluctuations may impact accuracy however.**

In laboratory calibrations, we do not observe any significant correlation of measured concentrations with sample pressure over this small range.

**Page 6 line 5 & Figure 2: I am unclear on this linear transforming one instrument to the other. More clarity is needed. I also am concerned that this may not be appropriate for CO2 and CH4. When I look at figure 2 I get greatly concerned as the variation from flight to flight is actually very significant for gases that we care about fractions of a ppm (ppb for CH4). I also worry about inflight variations. We need more information on the validation of the GHG obs.**

We have added the following text to Sect. 2.4.

*The transformation requires several operations, including 1) averaging dry mixing ratios to a common 1-Hz time-base, 2) smoothing LGR data to match the slower cell throughput of the Picarro, 3) time-lagging the LGR (typically < 2 s) to optimize correlation with the Picarro, 4) calculation of fit coefficients for an ordinary least-squares fit (LGR = m\*PIC + b), and 5) correction of LGR dry mixing ratios using the*

*fit parameters. This procedure, akin to performing a flight-by-flight span and intercept calibration correction, rectifies calibration errors and flight-to-flight drift that may occur in the LGRs under different operating conditions.*

As mentioned above, accuracy is not critical for airborne EC. We have also added the following to the end of Sect. 2.4.

*It is not currently feasible to calibrate the LGR systems in-flight due to high gas flow rates. As a result of our correction procedure, we estimate that the accuracy of LGR $CO_2$ and $CH_4$ is degraded by a factor of 3 and 4, respectively, compared to the Picarro (Table 1). It may be possible to expand the above correction method to account for potential in-flight variability in LGR accuracy; however, as discussed later, measurement accuracy is a negligible contributor to total flux uncertainty for greenhouse gases.*

**Section 2.5 (and figure 3): This section is somewhat of an aside. There is not other usage or discussion of this system.**

This system was part of the CARAFE payload and deserves mention. At present we have not been able to devote resources to making full use of this data, but we feel it is important to let readers know of its availability and discuss how it was acquired.

**Page 7 line8-9: This +-20 m requirement is very tight (as needed)- and will restrict the ability to use this data. Also, it should be made clear this is above ground and not asl. This makes it even harder to meet this requirement over terrain with any variability exceeding 20m. Further the 5 degree restriction is tough, but might it need to be tighter?**

This sentence specifically states that the altitude range is with reference to ground level. The limit of +/- 20 m was chosen for this particular set of flights based on observed variability in the data. It is not a hard limit for airborne EC. We have added the following statement to clarify:

*We note that this altitude window is based on observed variability in the 2016 and 2017 data-sets and is not a hard limit.*

We do not require flat terrain; for example, (Wolfe et al., 2015) reported fluxes over rolling terrain with ground altitudes of 200 – 350 m asl. Airborne EC is of course not possible over extremely rough terrain (mountains). We discuss this point in the revised introduction.

The 5-degree roll restriction is to ensure quality in the vertical wind data. The limit is based on inspection of the data and expert judgment from operators who have flown similar systems on multiple NASA aircraft. These data filtering procedures are standard among airborne EC groups (Misztal et al., 2014).

**Page 7 Line 22: The problem is undersampling of the PBL depth can lead to systematic biases, and this is a major problem.**

PBL depth variations have little bearing on derived fluxes and do not, to our knowledge, lead to systematic biases. We have added some text to clarify this point. Misztal et al. (2014) estimate a maximum uncertainty of 3% in isoprene surface fluxes resulting from +/- 100 m uncertainty in PBL depth, but they also arrive at their divergence correction differently from us.

**Page 11 Line 19: I'm confused, I had thought the authors earlier asserted 10Hz wasn't necessary, but here is seems this is a important error term.**

Median values for $SE_{RT}$ are 4-5%, which is roughly double that of $SE_{turb}$. We have modified the text here to be more specific. Though non-negligible, this error term is smaller than typical random errors. Moreover, it is systematic and thus correctable, as we discuss in revisions to Sect. 3.4.1 following Reviewer 1's suggestions.

**Page 14 line 18: This is disappointing. If all these flight hours are being used there should be more planning for multiple altitude legs for this type of validation.**

We have deleted this sentence as it is irrelevant to the discussion. Divergence corrections are a well-established aspect of airborne EC. We elected to allot most flight time to collecting near-surface observations to maximize statistical robustness in derived surface fluxes.

**Page 15 lines 7-8: And these large uncertainties are a major problem for the approach. With relative uncertainties that push to 100% the utility of the technique is degraded.**

The quoted uncertainties represent a full range across a wide distribution with a long tail. We have changed the error discussion throughout to consistently use interquartile ranges, and this is stated at the beginning of Sect. 3.4. We have also modified Fig. 7 and added text (Sect. 3.4.4 and 3.4.5) to clarify total errors in surface fluxes and the effects of averaging. For example, the interquartile range for uncertainty in CO2 fluxes is $18 - 30\%$ for leg averages and $40 - 90\%$ for 2 km averages. This is comparable to uncertainties reported other airborne flux studies (Vaughan et al., 2016).

**Page 16 line 15-16: This is fair, but it means this manuscript has not established the utility of this approach.**

As discussed above, we believe that numerous previous studies have established the utility of airborne EC.

**Conclusions: I take issue with much of how the conclusion is written (see major comment above). This would be better served to summarize the aircraft system and payload, and then highlight the challenges in the EC approach and the resulting expected uncertainties.**

We have added several sentences to the opening paragraph discussing the payload and uncertainties. Regarding the remainder of this section, this seems to be the appropriate place to discuss potential uses of this dataset and future potential applications of the technique in general.

**Page 17 Lines 29-30: It has not been established that this approach should be a part of a standard toolbox.**

Respectfully, we disagree. Numerous previous studies have highlighted the value of such measurements, as detailed in the introduction. The atmospheric chemistry community has made significant strides with this technique over the last few years, and the carbon/biosphere community has benefited from it in the past. As we state, acquisition of the data is only the first step. We are now developing tools to analyze this data in a scientific context.

At any rate, we have modified this sentence following the reviewer's previous comment, as follows:

*The NASA CARAFE project aims to incorporate eddy covariance fluxes as a standard component of the airborne science toolbox*

**Page 18 Lines 7-8: This isn't a new vector – as stated by authors the approach is old, and has been applied to Carbon before. The utility has never really been established, particularly given the limitations, and that is why it has hardly been adopted.**

We have changed the wording here. Please also see our response to the comment above.

**Page 18 Line 17: This does not clearly follow.**

We have changed "would" to "may" and added several references.

**Table 1: Methane and CO2 should be shown in ppb and ppm respectively (not fractional uncertainty).**

Fixed.

**Figure 1: The authors should indicate which flight legs are actually of use for EC on this plot – showing all the flight legs is misleading. 2016 data was deemed not useable, so should not be shown.**

We have changed the map to include only flux legs. As discussed above we have not deemed the 2016 data "unusable," thus we have left this information on the map.

**Figure 2: These are worryingly large to me. Also, this should show 2017 data as the authors don't use the 2016 flights.**

The quality of concentration measurements for 2016 and 2017 are comparable, it is only the fluxes that bear the systematic error from turbulence sampling. We have added the 2017 version of this figure to the appendix and added a reference to this figure in the text.

**Figure 7: This plot is sobering, and the log scale relative error brings into question the utility for CO2 and CH4.**

Please see our response to the comment on Page 15 above. The bulk of the distribution for total errors is not unreasonable.

**References**

Chen., H., Winderlich, J., Gerbig, C., Hoefer, A., Rella, C. W., Crosson, E. R., Van Pelt, A. D., Steinbach, J., Kolle, O., Beck, V., Daube, B. C., Bottlieb, E. W., Chow, V. Y., Gantoni, G. W., and Wofsy, S. C.: High-accuracy continuous airborne measurements of greenhouse gases ($CO_2$ and $CH_4$) using the cavity ring-down spectroscopy (CRDS) technique, Atmos. Meas. Tech., 3, 375-386, 2010.

Karion, A., Sweeney, C., Wolter, S., Newberger, T., Chen, H., Andrews, A., Kofler, J., Neff, D., and Tans, P.: Long-term greenhouse gas measurements from aircraft, Atmos. Meas. Tech., 6, 511-526, 2013.

Misztal, P. K., Karl, T., Weber, R., Jonsson, H. H., Guenther, A. B., and Goldstein, A. H.: Airborne flux measurements of biogenic volatile organic compounds over California, Atmos. Chem. Phys., 14, 10631-10647, 2014.

Ren, X., Hall, D. L., Vinciguerra, T., Benish, S. E., Stratton, P. R., Ahn, D., Hansford, J. R., Cohen, M. D., Sahu, S., He, H., Grimes, C., Salawitch, R. J., Ehrman, S. H., and Dickerson, R. R.: Retraction: Methane Emissions From the Marcellus Shale in Southwestern Pennsylvania and Northern West Virginia Based on Airborne Measurements, J. Geophys. Res. Atmos., doi: 10.1002/jgrd.54397, 2018. n/a-n/a, 2018.

Vaughan, A. R., Lee, J. D., Misztal, P. K., Metzger, S., Shaw, M. D., Lewis, A. C., Purvis, R. M., Carslaw, D. C., Goldstein, A. H., Hewitt, C. N., Davison, B., Beevers, S. D., and Karl, T. G.: Spatially resolved flux measurements of NOx from London suggest significantly higher emissions than predicted by inventories, Faraday Disc., 189, 455-472, 2016.

Wolfe, G. M., Hanisco, T. F., Arkinson, H. L., Bui, T. P., Crounse, J. D., Dean-Day, J., Goldstein, A., Guenther, A., Hall, S. R., Huey, G., Jacob, D. J., Karl, T., Kim, P. S., Liu, X., Marvin, M. R., Mikoviny, T., Misztal, P. K., Nguyen, T. B., Peischl, J., Pollack, I., Ryerson, T., St Clair, J. M., Teng, A., Travis, K. R., Ullmann, K., Wennberg, P. O., and Wisthaler, A.: Quantifying sources and sinks of reactive gases in the lower atmosphere using airborne flux observations, Geophys. Res. Lett., 42, 8231-8240, 2015.